# TAPVid-360: Tracking Any Point in 360 from Narrow Field of View Video

**Finlay G. C. Hudson**
Department of Computer Science
University of York
York, United Kingdom
finlay.gc.hudson@york.ac.uk

**James A. D. Gardner**
Department of Computer Science
University of York
York, United Kingdom
james.gardner@york.ac.uk

**William A. P. Smith**
Department of Computer Science
University of York
York, United Kingdom
william.smith@york.ac.uk

## Abstract

Humans excel at constructing panoramic mental models of their surroundings, maintaining object permanence and inferring scene structure beyond visible regions. In contrast, current artificial vision systems struggle with persistent, panoramic understanding, often processing scenes egocentrically on a frame-by-frame basis. This limitation is pronounced in the Track Any Point (TAP) task, where existing methods fail to track 2D points outside the field of view. To address this, we introduce TAPVid-360, a novel task that requires predicting the 3D direction to queried scene points across a video sequence, even when far outside the narrow field of view of the observed video. This task fosters learning allocentric scene representations without needing dynamic 4D ground truth scene models for training. Instead, we exploit 360 videos as a source of supervision, resampling them into narrow field-of-view perspectives while computing ground truth directions by tracking points across the full panorama using a 2D pipeline. We introduce a new dataset and benchmark, TAPVid360-10k comprising 10k perspective videos with ground truth directional point tracking. Our baseline adapts CoTracker v3 to predict per-point rotations for direction updates, outperforming existing TAP and TAPVid 3D methods. Project page: https://finlay-hudson.github.io/tapvid360

## 1 Introduction

Humans possess a remarkable ability to construct panoramic internal representations of space – moment-by-moment models that mentally complete the full sphere of surrounding information, even when only a fraction is currently visible. Coupled with object permanence, spatial mapping, and predictive scene completion, these cognitive mechanisms allow us to infer the full structure of a scene, updating our mental model dynamically as new information arrives. For example, we can sit down on a chair without looking behind us, this capability is based on maintaining a mental model that understands the chair remains despite being outside of direct view.

In contrast, current artificial vision systems struggle with this kind of persistent, panoramic scene understanding, as they often operate in an egocentric, frame-by-frame manner with limited memory for unseen regions. This is particularly significant in the context of the Track Any Point (TAP) task [1–6]. The goal of the TAP task is to track a set of 2D points through a video $V = (I_t)_{t=1}^{T}$ comprised of $T$

39th Conference on Neural Information Processing Systems (NeurIPS 2025) Track on Datasets and Benchmarks.

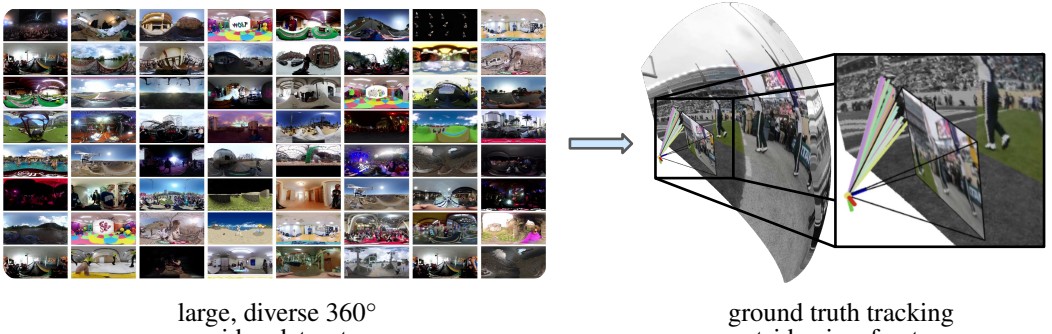

large, diverse 360°
video dataset

ground truth tracking
outside view frustum

Figure 1: Overview: the TAPVid-360 task (right) is to track the direction of a set of query points in the camera coordinate system of each frame of a narrow field of view video. We generate training data and a benchmark evaluation for this task by curating a large, diverse dataset of 360 videos. We track groups of points using a 2D segmentation and TAP pipeline and convert to directions. We then render a virtual narrow field of view perspective video by resampling the 360 video where the tracked directions may be (significantly) outside the field of view of the video.

RGB frames $I_t \in \mathbb{R}^{3 \times H \times W}$. Concretely, given a set of $N$ query points $(P_1^i)_{k=1}^N$, the goal is to predict the set of tracks for the points through all frames, $P_t^k = (i_t^k, j_t^k) \in \mathbb{R}^2, t = 1, \dots, T, k = 1, \dots, N$. Some methods additionally predict a binary visibility flag to indicate whether the point is occluded.

The original 2D formulation of the TAP task lacks panoramic scene understanding. Specifically, there is no notion of object permanence when an object leaves the field of view. Some methods are explicitly trained to track points slightly outside the field of view of the image but still within the image plane, meaning $(i_t^k, j_t^k)$ may not lie in $\{1, \dots, W\} \times \{1, \dots, H\}$. Any method that cannot predict point tracks outside the field of view of the image can be trivially extended to do so by padding each frame of the video with a border region. However, this representation cannot handle points that move far outside the original field of view, particularly any point in the back hemisphere (i.e. $> 90°$ from the view direction). Hence, most methods do not meaningfully track points once they are no longer visible and must resume tracking as a re-ID problem when they return.

A potential solution to this problem was the introduction of the TAPVid-3D [3] task. Here, the query points are still provided as 2D pixel coordinates but, in one possible formulation, the task is to predict a 3D trajectory $P_t^k = (u_t^k, v_t^k, w_t^k) \in \mathbb{R}^3$. Where the 3D points are either in a world coordinate frame that remains fixed or in the coordinate system of the camera at the corresponding frame. In principle, this alleviates the representation problem for out-of-view points since any 3D location can be represented – similarly to SLAM [7]. The downside of such a representation is the difficulty of acquiring training data. Complete 3D models for a possibly dynamic scene are required at every time instant. For this reason, the TAPVid-3D benchmark and most methods operate in a 2.5D representation where points are tracked in image space along with their depth, i.e. $P_t^k = (i_t^k, j_t^i, w_t^k) \in \mathbb{R}^3$, such that their 3D location in camera coordinates can be derived via projection using the camera intrinsics. This means that these methods still suffer the same problem as 2D TAP methods with regards to persistent tracking of points outside the field of view.

The recent availability of consumer-grade 360° cameras has led to an explosion of panoramic video data, providing an unprecedented opportunity to train models to develop allocentric scene representations – world-centric models of the environment that persist beyond momentary views. We exploit large-scale datasets of 360° videos, captured in diverse real-world conditions, as a rich source of supervisory signals for learning how spatial information unfolds beyond the boundaries of a given viewport. 360 video offers many unique advantages over traditional 2D perspective video. Namely, it offers a complete 360 view of the entire world around the camera. A lack of boundaries or borders to the view frustum means objects never leave frame. It provides a richer understanding of spatial layouts and scene dynamics.

In this context, we introduce the TAPVid-360 task. Here, given query points as pixel coordinates in the first frame, the goal is to track the 3D *direction* (in the camera coordinate frame) to the scene

point corresponding to the query point. Intuitively, we are asking the model to persistently predict in which direction a point is but (unlike TAPVid 3D) not its distance. This corresponds to a human being able to approximately point to where they believe a chair is behind them without knowing exactly how far away it is.

Concretely, given a narrow field of view, perspective video and query pixels for frame 1: $Q^k = (i^k, j^k), k = 1, \ldots, N$ (as for the TAPVid task), the goal is to predict directions in the form of unit vectors for each point across the sequence: $D_t^k = (x_t^k, y_t^k, z_t^k), t = 1, \ldots, T, k = 1, \ldots, N$ with $\|D_t^k\| = 1$, *even when the point has left the field of view*. We believe that this task provides a useful learning objective for many downstream tasks. For points that leave the field of view, the model is required to reason about egomotion and, for any camera motion other than pure rotation, *the 3D location of the point*. For dynamic points, it must additionally extrapolate the dynamic motion, possibly including an understanding of physical laws, for the unobserved period. Since directions must be predicted for every frame, it imposes object permanence as a hard constraint.

In this paper, our key contribution is to show how to construct a TAPVid-360 dataset *without requiring 3D ground truth*. Instead, we generate perspective, narrow field of view videos by resampling 360° videos while using a novel 2D point tracking pipeline on the complete 360° video to provide ground truth directions. We propose a baseline method to tackle this task by modifying and fine-tuning CoTracker v3 [5] to predict rotations for each query point/frame that update the predicted direction for that point from one frame to the next. We show that this outperforms existing TAPVid and TAPVid-3D methods when evaluated for the TAPVid-360 task.

## 2   Related Work

**Track Any Point**   Establishing correspondences across video frames is a fundamental problem in computer vision. Traditional optical flow methods focus on computing dense correspondences between consecutive frames. Early techniques relied on variational approaches and hand-crafted features [8, 9], but the advent of deep learning brought significant advancements with architectures like FlowNet [10] and RAFT [11]. However, these methods often struggle with long sequences, occlusions, and significant appearance changes, limiting their effectiveness in complex real-world scenarios.

To address these shortcomings, recent research has shifted toward Tracking Any Point (TAP), first introduced in [12], which emphasises long-term tracking capable of handling occlusions and appearance variations. The majority of current TAP models track in 2D [1, 2, 4–6], though some recent works extend the tracking problem to 3D [13, 14, 3]. However, no existing 3D method continues to estimate tracks for points that have left the camera's view frustum. A major cause of this limitation is the scarcity of suitable ground truth training data. The majority of data comes from synthetic dataset [15, 16], driving scenarios [17, 18], or complex 3D capture domes [19], which do not offer sufficient scale or diversity for training models to handle such out-of-view trajectories.

**360° Video for Scalable Supervision**   Addressing the challenge of data scarcity for 3D tracking necessitates novel data sources. The rise of consumer-grade 360° cameras has resulted in a vast source of data that remains largely untapped. While some recent works have begun to leverage this data, their focus has primarily been on generative tasks. For instance, [20] presented a dataset of one million 360° YouTube videos for training novel view synthesis models, and [21] utilised this dataset to finetune a video diffusion model for 360° outpainting from perspective inputs. In this work, we demonstrate that 360° video not only offers the unique capability to track objects continuously as they move within the full spherical field of view, but also enables the resampling of arbitrary perspective camera trajectories. This provides a diverse and readily scalable source of supervision for TAP models.

## 3   Dataset Generation

To train models in this task we require 360° video data, filtered for quality and dynamic content. From such a dataset we can extract paired perspective crops and ground-truth camera-relative direction tracks to objects within the scene. To that end, we first curate a high-quality dataset of 360° videos and then process these to provide pseudo-ground-truth tracks.

### 3.1 Data Curation

We start with the 360-1M dataset [20], a dataset of approximately 1 million YouTube links for 360° videos. However, many of the links provided point to non-360° videos or videos that are incorrectly formatted for our use case. We follow similar filtering methods to [21, 22] to filter the data before processing for point tracking. We store our 360° videos as 2D videos under equirectangular projection.

**Coarse Filtering** We keep only videos with greater than 15 likes, around 100k videos, as a baseline for quality. We download in the highest quality available using yt-dlp [23]. We then remove any videos that don't contain *side-data-list* in their metadata (metadata indicating to media players to play this content in 360° format). If the *side-data-list* has the value *top and bottom*, we crop and scale to a 2:1 aspect ratio the top half of the video. This filters for videos that

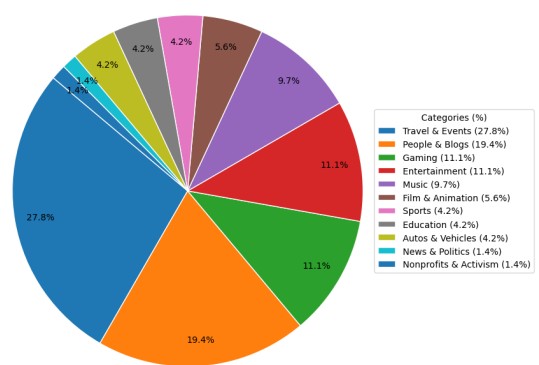

Figure 2: Distribution of categories in our filtered dataset. The largest majority comes from Travel and Events with 27.8%.

should not have been in the 360-1M dataset; however, many videos are incorrectly labelled or are 360° formatted videos containing perspective videos or images projected onto a sphere.

To remove these we therefore apply the following coarse-filtering process:

- **Perspective / Poster Detection** Videos are grayscaled and binarised using adaptive thresholding. The bounding box of the largest external contour, representing the main content, is determined. If this content area is significantly smaller than the frame and the surrounding regions are black borders the video is flagged as a poster and removed.

- **Scene Dynamics** Frames are sampled at random intervals, and pixel variance is calculated. Static videos with minimal inter-frame variation are removed.

- **Formatting** LPIPS [24] between the left and right halves is computed to filter 180° formatted videos and between the top and bottom halves to filter 360 videos with incorrect metadata. 180° videos are removed and the *top and bottom* are cropped and scaled to 2:1 aspect ratio as before.

- **Seams** Since equirectangular video should be continuous at the edges, discontinuities at the wrap-around seam (left and right edges) of the equirectangular video indicate a non-360° video. Thin vertical strips are extracted from the leftmost and rightmost edges of the frame, and normalised cross-correlation is computed between the crops. This yields a similarity score between -1 and 1. If the similarity score falls below a specified threshold, it indicates a mismatch between the edges, implying a visible seam.

These metrics are averaged over ten evenly spaced frames throughout the video.

**Fine Filtering** After coarse filtering, we split the videos into 10-second clips using FFmpeg [25]. These require further filtering as it is still possible for clips to contain minimal dynamic content, to have scene changes partway through and for watermarks to be placed over the video.

We therefore use the following fine-filtering process:

- **Optical Flow** Calculate the average magnitude of sparse optical flow vectors between every other frame and remove videos below a threshold.

- **Scene Detection** Clips are run through PySceneDetect [26] to identify scene changes either with harsh cuts or through fades. Clips that contain scene changes are removed.

- **Watermarks** A LAION Watermark detection network [27] is used to identify watermarking and remove clips above a confidence threshold.

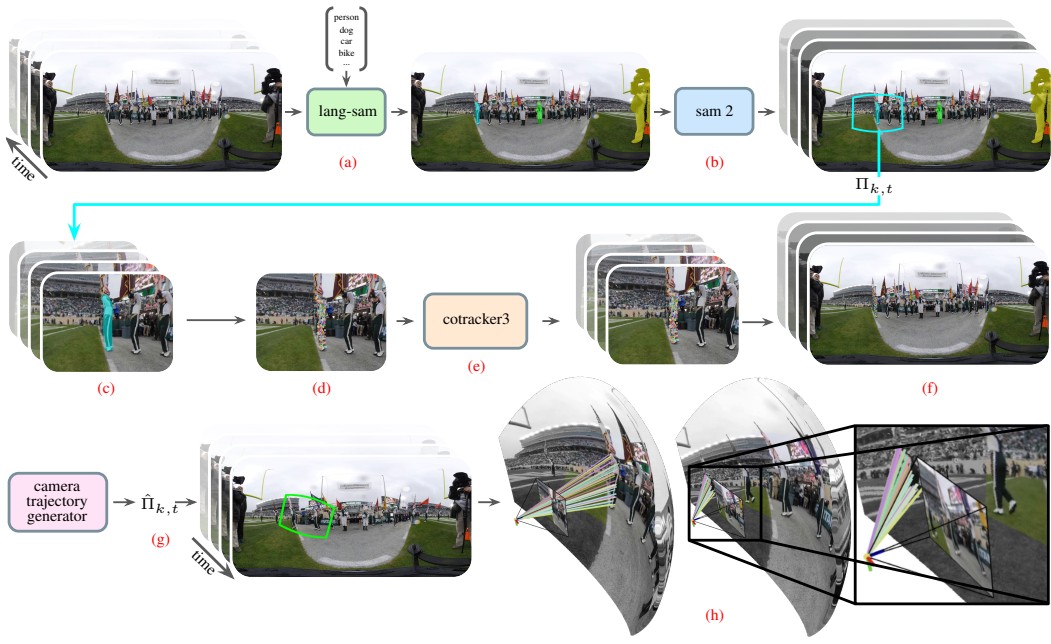

Figure 3: Overview of our data generation pipeline (zoom for detail). We first use Lang-SAM [28] on the first frame of our pre-filtered 360° video to segment dynamic objects. SAM2 is then used to distribute these masks to the full video. We sample 2D perspective videos following each object mask and use CoTracker3 [5] to track the object through the perspective video. We then transform these tracks back onto the 360° video. Finally we sample novel camera trajectories and use these new 2D perspective videos and ground truth 3D directions as training data for our model.

This results in around 130k 10-second clips that are correctly formatted and contain good dynamic content or camera motions.

## 3.2 Ground Truth Directional Point Tracks from 360° Video

Given our set of filtered clips, we generate training and evaluation datasets comprising paired perspective videos and camera-relative, pseudo-ground-truth, unit-vector tracks for dynamic objects. An input 360° video in equirectangular format is denoted as $V_{eq} = (\mathcal{I}_t)_{t=1}^T$, representing a sequence of $T$ frames where each frame $\mathcal{I}_t \in \mathbb{R}^{3 \times H_{eq} \times W_{eq}}$. We first subsample $V_{eq}$ to a new sequence length, of $T = 32$ frames, to increase the likelihood of capturing salient dynamics within the clip. Lang-SAM [28] is applied to the initial equirectangular frame $\mathcal{I}_1$, Figure 3 (a), using a predefined set of object classes typically exhibiting dynamic behavior e.g. person, bicycle, dog, car. See supplementary for the full list. This yields an initial set of instance segmentation masks for frame $\mathcal{I}_1$, $M_{init} = \{M_{i,j}\}_{j=1}^{N_{det}}$ for $N_{det}$ detected instances. We retain masks whose confidence scores exceed a predefined threshold $\tau_{conf}$ and select the top-$K$ scoring masks from this filtered subset. These $K$ masks $\{M'_{1,k}\}_{k=1}^K$, serve as initialisations for SAM2 [29]. SAM2 is subsequently run on the complete $T$ frame equirectangular video $V_{eq}$ to propagate these masks, Figure 3 (b), thereby obtaining instance segmentation masks for each of the $K$ objects across all frames. If SAM2 fails to maintain a consistent mask for an instance throughout the video's duration, that instance is excluded. This procedure results in a final set of equirectangular segmentation masks $SM_{eq} = \{S_{t,k} \mid t \in \{1, \ldots, T\}, k \in \{1, \ldots, K'\}\}$ where $K' \leq K$ is the number of successfully tracked instances and each $S_{t,k} \in \{0, 1\}^{H_{eq} \times W_{eq}}$.

**Perspective Projections and Point Tracks**   For each of the $K'$ successfully tracked instances, we now generate corresponding perspective views and 2D point tracks, which will subsequently be used to derive pseudo-ground-truth unit vectors. For each instance $k \in \{1, \ldots, K'\}$, we generate a sequence of time-varying camera projection parameters, Figure 3 (c). Specifically, for each frame

$t \in \{1, ..., T\}$ of the intended perspective clip, we sample an extrinsic rotation matrix $\mathbf{R}_{k,t} \in SO(3)$ and an intrinsic camera matrix $\mathbf{K}_{k,t} \in \mathbb{R}^{3 \times 3}$ (which defines the FOV for that frame). These parameters define a time-dependent perspective camera transformation $\Pi_{k,t}(\cdot; \mathbf{R}_{k,t}, \mathbf{K}_{k,t})$.

Using this sequence of transformations, we render a perspective video $V_{persp,k} = (\mathcal{I}'_{t,k})_{t=1}^{T}$ from the equirectangular video $V_{eq}$, where each frame $\mathcal{I}'_{t,k} \in \mathbb{R}^{3 \times H_{persp} \times W_{persp}}$ is rendered using its corresponding $\mathbf{R}_{k,t}$ and $\mathbf{K}_{k,t}$. Similarly, we project the instance's equirectangular segmentation masks $\{S_{t,k}\}_{t=1}^{T}$ to obtain a sequence of perspective masks $SM_{persp,k} = (S'_{t,k})_{t=1}^{T}$, where $S'_{t,k} \in \{0,1\}^{H_{persp} \times W_{persp}}$. **The sequence of projections $\Pi_{k,t}$ is chosen such that the instance $k$ is centred within the perspective view, guided by its mask sequence.**

For the initial frame $\mathcal{I}'_{1,k}$ of the $k$-th perspective video, we sample as set of $N_q$ query pixels coordinates $\mathcal{P}_{1,k}\{(i_q, j_q)\}_{q=1}^{N_q}$ from within the corresponding projected segmentation mask $S'_{1,k}$, Figure 3 (d). These 2D points, associated with the first frame (time index $t = 1$) are then formulated as a set of spatio-temporal queries $\mathcal{Q}_k = \{(1, u_j, v_j)\}_{j=1}^{N_q}$. This set of $\mathcal{Q}_k$ is provided as input to CoTracker3 [5], Figure 3 (e), to obtain 2D point tracks $\mathbf{p}_{t,q} = \{(i_{t,q}, j_{t,q})\}_{t=1,...,T; q=1,...,N_q}$ within the perspective video $V_{persp,k}$.

The set of tracks is further refined by filtering based on cumulative 2D displacement. For each track $q$, represented by the sequence of image-plane coordinates $\mathbf{p}_{t,q} = (i_{t,q}, j_{t,q})$ for frames $t = 1, \ldots, T$, we calculate its total path length. This cumulative length, $L_q$, is given by the sum of Euclidean distances between temporally consecutive points:

$$L_q = \sum_{t=1}^{T-1} \|\mathbf{p}_{t+1,q} - \mathbf{p}_{t,q}\|_2$$

Tracks are retained only if $L_q$ exceeds a specified cumulative length threshold, $L_{thresh}$. This selection keeps tracks that exhibit dynamic object motions, effectively removing tracks from static, distant or minimally moving points.

We now transform the filtered 2D perspective tracks into direction vectors, Figure 3 (f). Given our 2D perspective point tracks $\mathcal{P} = (i_{t,q}, j_{t,q}) \in \mathbb{R}^2, t = 1, ...., T, q = 1, ...., N_q$, the corresponding unit vector in camera coordinates can be computed as follows:

For each tracked 2D point $\mathcal{P} = (i_{t,q}, j_{t,q})$ from the $q$-th track at frame $t$ in the $k$-th perspective video, we first represent it in homogenous coordinates as $\tilde{\mathcal{P}} = (i_{t,q}, j_{t,q}, 1)$. This is transformed into a 3D direction vector in the camera's coordinate system as:

$$\hat{\mathbf{d}}_{cam,t,q} = \frac{\mathbf{v}_{cam,t,q}}{\|\mathbf{v}_{cam,t,q}\|_2}, \quad \mathbf{v}_{cam,t,q} = \mathbf{K}_{k,t}^{-1} \tilde{\mathcal{P}}_{t,q}. \tag{1}$$

**Sampling New Perspective Crops**  Now that we have $\hat{\mathbf{d}}_{cam,t,q}$, our pseudo-ground-truth unit-vector representing the direction from the camera centre to the point $(i_{t,q}, j_{t,q})$ in the camera's 3D coordinate system for that specific frame $t$ and instance $k$, we can generate training examples by resampling new perspective camera trajectories from the equirectangular video. As before for each instance $k \in \{1, \ldots, K'\}$, we generate a sequence of time-varying camera projection parameters, Figure 3 (g). Specifically, for each frame $t \in \{1, ..., T\}$ of $V_{eq}$ we sample a time-dependent perspective camera transformation $\hat{\Pi}_{k,t}(\cdot; \hat{\mathbf{R}}_{k,t}, \hat{\mathbf{K}}_{k,t})$. However the sampling of this perspective transform is now chosen randomly from a set of predefined camera motions functions, *static (original camera motion)*, *spin_(x,y,z)*, *spiral*, *simulated human*, *random* and *btf*. See supplementary for detailed descriptions of these different sampling methods. The result is 5k training and 10k test samples. All containing dynamic camera motions with ground truth tracks for objects that can leave the view frustum.

### 3.3   Dataset Statistics

Table 1 presents a comparison of our TAPVid360-10k validation set with existing TAP datasets, spanning both 2D and 3D modalities. Notably, despite representing only the validation portion of our dataset, TAPVid360-10k exhibits strong coverage and diversity. Moreover, our data generation pipeline, presented in Section 3, is capable of producing significantly larger-scale datasets with

similar characteristics. In addition to these aggregate metrics, Table 2 highlights the distribution of annotated points that fall within versus outside the perspective camera's field of view, illustrating the dataset's breadth across different visibility conditions.

## 4 CoTracker360: A TAPVid-360 Baseline

As a baseline model, we modify the recent CoTracker3 [4] method to predict directions instead of point estimates. The original CoTracker3 predicts point displacements relative to the first frame. This is an easier representation for the model to reason about compared to directly regressing absolute point position at each frame. We follow the same approach except that we apply a *rotation* to the direction at the first frame. Accordingly, we replace the last layer of the CoTracker3 decoder with a linear layer with 9 outputs and linear activation. We reshape this to a $3 \times 3$ matrix and project to the closest rotation matrix using special orthogonal Procrustes orthonormalization [30].

To produce output, we convert the initial query point positions from pixel coordinates to directions using the intrinsic parameters of the camera. These unit vector directions are then rotated using the rotation matrices predicted for this point for each frame. We supervise the direction predictions using Huber loss (we experimented with angular error but found this to be less stable). We initialise with the pretrained CoTracker3 offline weights and finetune with a training dataset created using the data generation approach described in Section 3.2. We create 5k additional perspective video clips. These clips are distinct from those in the TAPVid360-10k dataset to avoid any overlap. We do not supervise the CoTracker3 confidence and visibility outputs during training. The model is trained for 120 epochs using the Adam optimizer with a learning rate of $1e-4$. Training is performed on a single NVIDIA A40 GPU with a batch size of 8. Due to memory constraints, we are limited in the number of query points and frames used during training; we choose 32 query points across 32 frames. We refer to this trained model as CoTracker360.

## 5 Evaluation

To establish a baseline for TAPVid-360, in addition to our proposed CoTracker360, we evaluate several state-of-the-art tracking approaches, including both 2D and 3D methods. For 2D tracking, we consider TAPIR [2], BootsTAPIR [31], and CoTracker3 [4], while for 3D tracking, we utilise SpatialTracker [13]. To align these methods with the dataset, which represents motion as unit vector directions, we convert their pixel-space outputs accordingly using the known camera intrinsics. We run evaluation using 256 query points and 32 frame clips.

**Metrics** To evaluate our benchmark, we adapt a widely used metric within TAP frameworks [1, 3, 4], namely $< \delta^x_{\mathrm{avg}}$, which measures the fraction of predicted points that lie within a given threshold of the ground truth. However, since our setting involves directional vectors rather than purely $(x, y)$ coordinates, we replace the pixel-based distance with an angular threshold, expressed in terms of angle per pixel. Based on the field of view of our dataset, a movement of a single

| Dataset | # Videos | # Clips | # Objects | Avg Trajectories Per Clip | Real/Sim | FPS | Data Type |
|---------|----------|---------|-----------|---------------------------|----------|-----|-----------|
| TAPVid-RGB-Stacking | 50 | 250 | / | 30 | Sim | 25 | 2D |
| RoboTAP | 265 | / | / | 44 | Real | / | 2D |
| TAPVid-Kinetics | 1,189 | / | / | 26.3 | Real | 25 | 2D |
| TAPVid-KUBRIC | 38,325 | / | / | flexible | Sim | 25 | 2D |
| TAPVid-3D | 2828 | 4569 | / | 50-1024 | Real | 10-30 | 3D |
| TAPVid360-10k | 4772 | 4772 | 10000 | 256 | Real | 3.75-30 | 360 |

Table 1: Comparison of dataset metrics.

| Dataset | # Points Within Frame | # Points Out of Frame |
|---------|----------------------|----------------------|
| TAPVid360-10k | 36.28M | 45.64M |

Table 2: Showing the number of points located within the visible frame versus those outside the frame boundaries.

| Method | $< \delta^x_{\mathrm{avg\,all}} \uparrow$ | $< \delta^x_{\mathrm{avg\,if}} \uparrow$ | $< \delta^x_{\mathrm{avg\,oof}} \uparrow$ | $AD^x_{\mathrm{avg\,all}} \downarrow$ | $AD^x_{\mathrm{avg\,if}} \downarrow$ | $AD^x_{\mathrm{avg\,oof}} \downarrow$ |
|---|---|---|---|---|---|---|
| TAPNext [6] | $0.0082_{\pm0.0070}$ | $0.0191_{\pm0.0157}$ | $0.0004_{\pm0.0009}$ | $51.9752_{\pm18.0770}$ | $36.5923_{\pm18.8858}$ | $62.4601_{\pm19.8415}$ |
| TAPIR [2] | $0.0106_{\pm0.0081}$ | $0.0251_{\pm0.0191}$ | $0.0003_{\pm0.0006}$ | $49.8086_{\pm17.9032}$ | $33.8824_{\pm20.9064}$ | $60.5154_{\pm17.5166}$ |
| BootsTAPIR [31] | $0.0126_{\pm0.0119}$ | $0.0293_{\pm0.0262}$ | $0.0005_{\pm0.0009}$ | $48.3582_{\pm17.9551}$ | $33.3154_{\pm20.5326}$ | $58.3841_{\pm17.9581}$ |
| TAPIP3D [32] | $\mathbf{0.2476}_{\pm0.1564}$ | $0.4698_{\pm0.2391}$ | $0.0850_{\pm0.1227}$ | $36.4412_{\pm25.8254}$ | $23.3191_{\pm21.8679}$ | $45.7951_{\pm31.5080}$ |
| SpatialTracker [13] | $0.2239_{\pm0.1052}$ | $0.4893_{\pm0.1946}$ | $0.0303_{\pm0.0439}$ | $38.8780_{\pm23.4462}$ | $22.1635_{\pm21.4754}$ | $50.4350_{\pm27.3025}$ |
| CoTracker3 (offline) [4] | $0.2435_{\pm0.0891}$ | $\mathbf{0.5588}_{\pm0.1574}$ | $0.0158_{\pm0.0224}$ | $37.4287_{\pm21.3983}$ | $17.6352_{\pm21.3299}$ | $50.9759_{\pm24.0512}$ |
| *Ours* (CoTracker360) | $0.2386_{\pm0.1141}$ | $0.4060_{\pm0.1355}$ | $\mathbf{0.1160}_{\pm0.1094}$ | $\mathbf{8.2749}_{\pm7.4466}$ | $\mathbf{3.9496}_{\pm4.4395}$ | $\mathbf{10.9829}_{\pm9.7290}$ |

Table 3: Comparison of tracking performance metrics.

pixel corresponds to $0.2755°$ (degrees per pixel, denoted as $px°$). Hence, our thresholds are set to $[1\,px°, 2\,px°, 4\,px°, 8\,px°, 16\,px°]$. The average value across all of these thresholded ranges is then computed to produce a single summary score. In addition to this, we also report the mean angular distance between each predicted point and its corresponding ground truth ($AD^x_{\mathrm{avg}}$). Given the challenging nature of the dataset, it is common for the deviation between predictions and ground truth to exceed the largest threshold defined by $< \delta^x$. In such cases, the thresholded accuracy alone may not fully capture the performance. Therefore, we include the average angular distance as a complementary metric that reflects the overall deviation, regardless of threshold. We split each of these metrics into *in-frame* (IF) and *out-of-frame* (OOF) subsets to separately evaluate a model's point tracking performance on each condition. In addition, we report results for all points. A key contribution of our work is this explicit evaluation of a model's ability to track points that move out-of-frame. This is an aspect that is often overlooked by existing benchmarks, which primarily focus on in-frame accuracy. At the same time, we ensure that in-frame performance is preserved and does not degrade significantly when models are trained or evaluated under this extended setting.

**Quantitative Results**   The results in Table 3 highlight a critical distinction between precision and reliability. While CoTracker3 and SpatialTracker show high precision on in-frame points ($< \delta^x_{\mathrm{avg\,if}}$), their performance is undermined by large error magnitudes on incorrect predictions, as reflected in their poor angular distance ($AD$) scores. Our method, CoTracker360, resolves this issue, and its superiority is most evident when tracking points out-of-frame. Our method achieves the best out-of-frame accuracy ($< \delta^x_{\mathrm{avg\,oof}}$) being 1.3x higher than the next-best baseline, TAPIP3D. Critically, it also reduces the corresponding angular distance error ($AD^x_{\mathrm{avg\,oof}}$), a greater than 4-fold reduction over TAPIP3D. The notable performance of both CoTracker360 and TAPIP3D (which uses 3D context) highlights that explicitly accounting for out-of-frame context is essential. This state-of-the-art error reduction demonstrates that CoTracker360 avoids the catastrophic failures of other models and most robustly handles the primary challenge of long-range, out-of-frame tracking.

**Qualitative Results**   In Figure 4 and 5 we show qualitative examples of the tracking results on sample videos from the TAPVid360-10k dataset. Frames are rendered as egocentric images within the equirectangular image frame. Query points are shown in frame 1 and the tracking results in subsequent frames. When the tracked object leaves the image frame, many comparison methods either lose the object entirely or remain stuck at the image border where the object left the image. CoTracker360 is able to continue making plausible estimates as to the direction of the points based on camera motion.

# 6   Conclusions

In this paper we have introduced the TAPVid-360 task and a scalable method to generate ground truth data. We have shown that a simple adaptation to an existing TAP model and finetuning on a small set of such data allows the model to track points well outside the field of view of the image and to continue to predict dynamic motion. The representation switch from image plane points to 3D directions abstracts the task away from the specific field of view of the image to a panoramic representation, akin to human allocentric representations, without requiring difficult-to-obtain ground 4D scenes models for supervision. We also argue that directional rather than absolute positional tracking is an easier task yet could still be used as pretraining for 3D related tasks.

**Limitations**   There are some limitations to both the dataset and proposed baseline method. The dataset is created using a fixed field of view. This allows us to test whether the TAPVid-360 task is

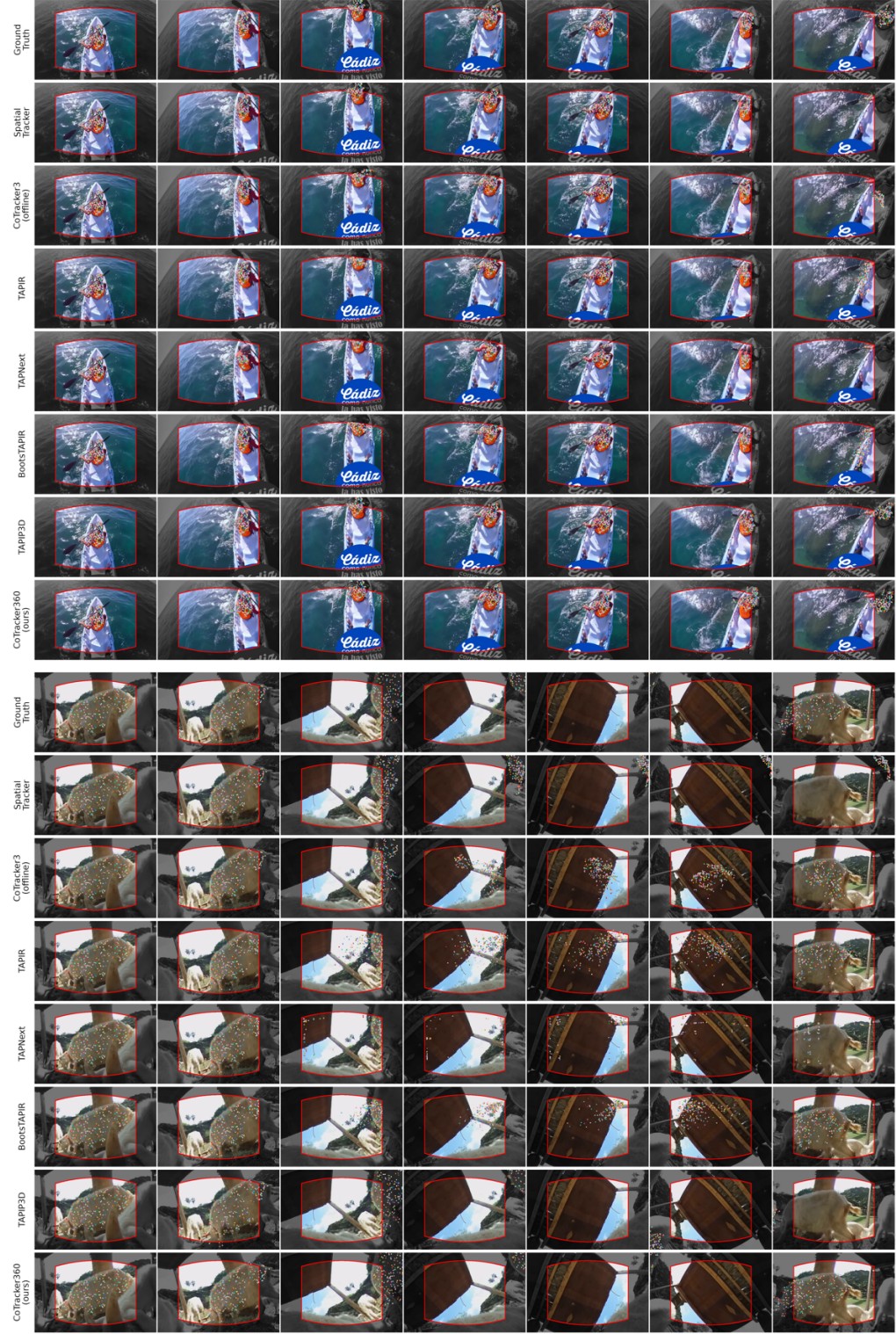

Figure 4: Direction tracks for each method for two videos (zoom for detail). Each frame is visualised as an egocentric perspective image within the equirectangular representation (greyed out regions are outside the field of view). The query points are shown in the first frame, the original video frames in the top row and ground truth tracks in the second row of each block.

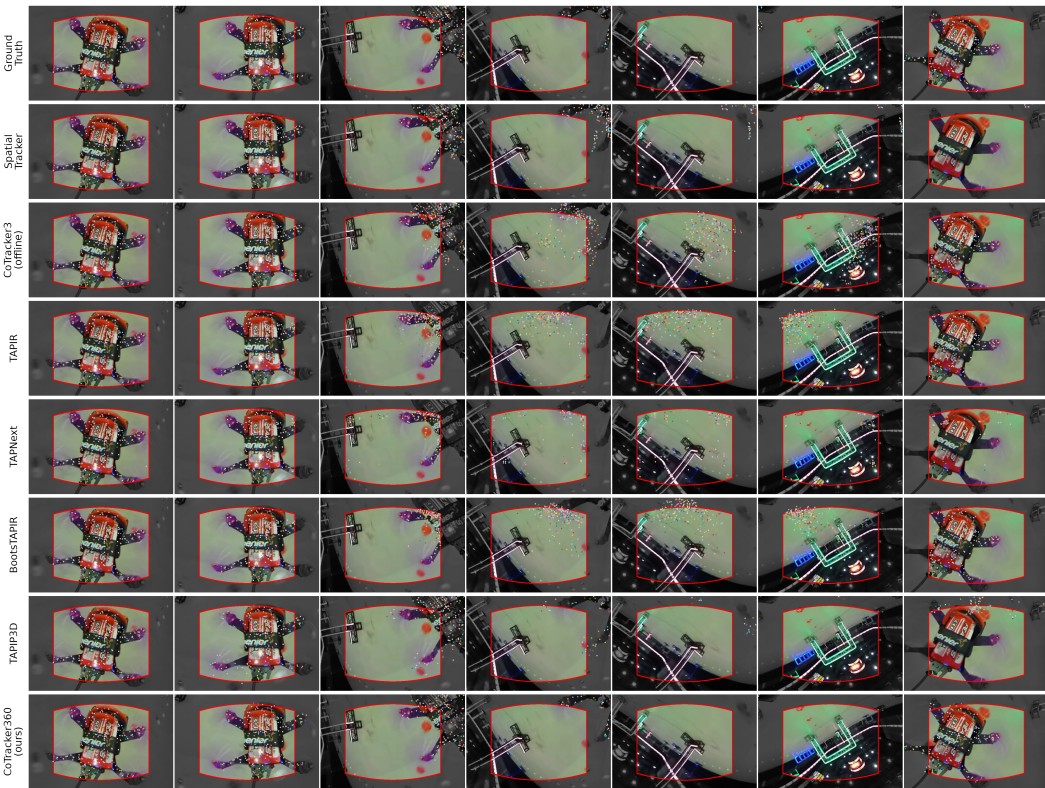

Figure 5: Further qualitative example of estimated direction tracks (see Figure 4 for details).

solvable in this restricted case but does not allow us to test sensitivity to field of view or dynamically changing field of view (i.e. zoom). Related to this, a limitation of the baseline method is that we can therefore use a fixed positional encoding per patch. To correctly allow the model to handle zoom, we would need to use a per-patch directional encoding based on the direction through the patch centre to encode field of field. The model would also benefit from representing uncertainty - i.e. a directional distribution. This would allow the model to be increasingly uncertain as points leave the field of view.

**Broader impacts**     Solving the TAPVid-360 task enables several significant downstream applications.

In robotics, maintaining a persistent, panoramic understanding of its surroundings would enable more robust active vision, allowing a robot to accurately reacquire objects that leave its field-of-view (FoV) by pointing its camera in the predicted direction. The predicted directional tracks can also serve as powerful priors for re-identification (re-ID) systems. When an object reappears, candidates that appear in locations requiring implausible motion can be rejected, improving tracking robustness when objects return to view.

The TAPVid-360 task requires that a model learns key cognitive abilities like object permanence and reasoning about unseen object dynamics. Given the difficulty of acquiring ground-truth temporal 3D tracks for dynamic scenes, our scalable data generation pipeline could be used to pretrain a model before being fine-tuned on smaller 3D datasets.

Finally, the directional tracks could be used as a conditioning signal for video generation models. This would help enforce temporal consistency and plausible object motion, mitigating common failure modes where objects are forgotten or hallucinated incorrectly after leaving and re-entering the camera's view.

We additionally acknowledge however that these same advancements could also have negative consequences if misused, potentially leading to more sophisticated surveillance capabilities that erode privacy, or contributing to the development of more effective autonomous weaponry with reduced human oversight.

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

Figure 6: The Graphical User Interface (GUI) for manual verification of object point tracks. Users can accept tracks to be passed to the next stage of the pipeline or reject them.

## A  Dataset Verification

To ensure high dataset fidelity, we employ a two-stage manual verification process. First, we validate the output of the point tracker on object-centred perspective crops, as detailed in *Perspective Projections and Point Tracks*. To streamline this step, we developed a verification GUI (Fig. 6) that enables an operator to efficiently accept or reject point tracks before they proceed to the camera motion emulation stage. Second, following camera emulation, a visualisation tool (Fig. 7) is used to confirm that the final data is plausible, accurate, and sufficiently diverse.

## B  Dynamic Object Classes

To query LangSAM, we select a curated list of object categories that are typically associated with dynamic behavior in real-world scenes. This strategy is aimed at maximizing the likelihood of capturing non-static points. The chosen categories are: person, bird, fish, insect, dog, cat, horse, snake, animal, car, bike, motorcycle, train, airplane, boat, ship, helicopter, submarine, rocket, bus, truck, robot, drone, conveyor belt, wind turbine, fan, clock hands, gears, ball, frisbee, pendulum, swing, yo-yo, kite, and shopping cart.

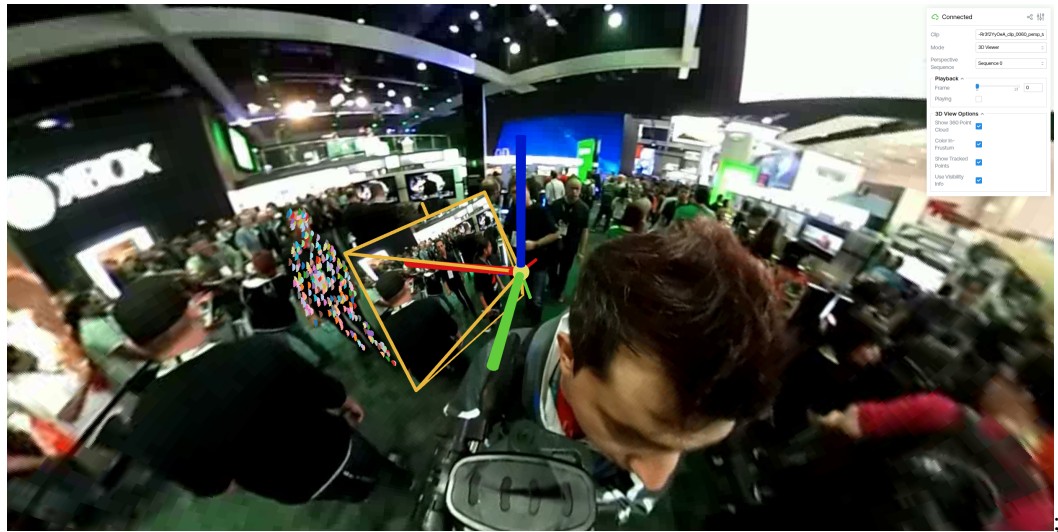

Figure 7: 3D Viser [33] based viewer used to check camera motion strategies and frustrum alignment. The 360 video wraps around the sampled perspective camera frustum shown here in orange. Local camera axis and world coordinate axis are also shown.

## C  Perspective Camera Sampling Methods

Given a specified number of frames $N$, we aim to generate a sequence of rotation matrices $\{\mathbf{R}_i\}_{i=0}^{N-1}$ simulating a form of camera motion. To achieve this we sample from a range of motion strategies, as well as employing an optional back-to-front (btf) strategy.

### C.1  Motion Strategies for Framewise Rotation

We define a motion strategy that governs the per-frame rotational deltas applied to an initial rotation matrix $\mathbf{R}_0 \in \mathrm{SO}(3)$, producing a sequence $\{\mathbf{R}_i\}_{i=0}^{N-1}$. The framewise deltas are determined by a selected motion type: `spiral`, `random`, `simulated human`, `static`, or `spin`. At each time step $i$, we compute pitch ($\alpha_i$), roll ($\beta_i$), and yaw ($\gamma_i$) angles, and update the rotation matrix accordingly:

$$\mathbf{R}_{i+1} = \mathbf{R}_i \cdot \mathbf{R}_{\text{update}}(\alpha_i, \beta_i, \gamma_i)$$

where $\mathbf{R}_{\text{update}} \in \mathrm{SO}(3)$ represents the combined rotation induced by pitch ($\alpha$), roll ($\beta$), and yaw ($\gamma$) angles, applied in a fixed axis order (e.g., XYZ).

Let $N$ be the total number of frames, and $\theta_{\min}, \theta_{\max} \in \mathbb{R}$ be the angular bounds.

- **Spiral Motion:**

$$\alpha_i = \gamma_i = \left( \frac{i(\theta_{\max} - \theta_{\min})}{N} \right) \bmod 360°, \quad \beta_i = 0$$

- **Random Motion:**

$$\alpha_i \sim \mathcal{U}_{\mathbb{Z}}[\theta_{\min}, \theta_{\max}], \quad \beta_i \sim \mathcal{U}_{\mathbb{Z}}[\theta_{\min}, \theta_{\max}], \quad \gamma_i \sim \mathcal{U}_{\mathbb{Z}}[\theta_{\min}, \theta_{\max}]$$

where $\mathcal{U}_{\mathbb{Z}}[a, b]$ denotes a uniform discrete distribution over integers in $[a, b]$.

- **Simulated Human Motion:** Inspired by [34], this strategy mimics natural human motion by combining a sinusoidal **oscillatory term** ($A_j \sin(\omega i)$), a **linear drift term** ($D_k i$), and a per-frame **noise term** ($\epsilon_j(i)$). The resulting rotational deltas are defined as:

$$\beta_i = A_r \sin(\omega i) + \epsilon_r(i) \tag{2}$$
$$\alpha_i = A_p \sin(\omega i) + D_p i + \epsilon_p(i) \tag{3}$$
$$\gamma_i = A_y \sin(\omega i) + D_y i + \epsilon_y(i) \tag{4}$$

where parameters are sampled once per sequence (unless noted) from:

$$A_j \sim \mathcal{U}(0, A_{j,\max}) \qquad \text{for axes } j \in \{r, p, y\}$$
$$D_k \sim \mathcal{U}(-D_{k,\max}, D_{k,\max}) \quad \text{for axes } k \in \{p, y\}$$
$$\omega \sim \mathcal{U}(\omega_{\min}, \omega_{\max})$$
$$\epsilon_j(i) \sim \mathcal{N}(0, \sigma_j^2) \qquad \text{(per frame)}$$

- **Static Motion (Original Camera Motion):**
$$\alpha_i = \beta_i = \gamma_i = 0$$

- **Spin Sequence (with Optional Noise):**
  In this strategy, the camera undergoes consistent rotation about a single axis $a \in \{x, y, z\}$, optionally perturbed by zero-mean noise. The nominal step size is:
$$\Delta\theta = \frac{360°}{N}$$

  A noise vector $\boldsymbol{\epsilon} \in \mathbb{R}^N$, where $\epsilon_i \sim \mathcal{U}(-\eta\Delta\theta, \eta\Delta\theta)$ for some noise ratio $\eta \in [0, 1]$, is used to perturb the steps:
$$\boldsymbol{\epsilon} \leftarrow \boldsymbol{\epsilon} - \frac{1}{N}\sum_{i=1}^{N}\epsilon_i$$

  The final rotation step per frame becomes:
$$\theta_i = \Delta\theta + \epsilon_i$$

  The corresponding Euler angle update is applied only along axis $a$, with others zeroed.

## C.2 Back-to-Front (btf) Sampling Strategy for Symmetric Motion Sequences

To simulate temporally coherent and reversible motion patterns, we introduce a rotation sampling procedure called back-to-front (btf). This approach constructs a symmetric camera motion sequence centered around a middle frame, ensuring the target object remains in view at both the start and end of the sequence, while allowing flexible motion in the intervening frames.

Given an initial sequence of rotation matrices $\{\mathbf{R}_i^{(0)}\}_{i=0}^{N-1}$, we define a subset of frames around the temporal midpoint to undergo smooth motion governed by a selected motion strategy (listed in Section C.1) . We then mirror the motion to maintain temporal symmetry.

**Midpoint-Based Symmetric Sampling**  Let $N$ be the total number of frames, and define:
$$m = \left\lfloor \frac{N}{2} \right\rfloor \quad \text{(temporal midpoint)}$$

We randomly select an even number of frames $k \in \{N_{\min}, \ldots, N_{\max}\}$ such that:
$$N_{\min} = \left\lfloor \frac{N}{2} \right\rfloor, \quad N_{\max} = N - 2 \cdot s$$

where $s = 2$ is a buffer to ensure the motion does not affect sequence boundaries. Let:
$$i_s = m - \frac{k}{2}, \quad i_e = i_s + k$$

**Forward and Reverse Rotation Generation**  Let $\{\mathbf{R}_i^{\text{init}}\}_{i=i_s}^{i_e}$ be the initial rotations for the selected region. We generate forward motion using a motion function $\mathcal{M}$ (e.g., `SpinSequence` along axis $z$):

$$\{\mathbf{R}_j^{\text{fwd}}\}_{j=0}^{k_f-1} = \mathcal{M}(k_f, \{\mathbf{R}_i^{\text{init}}\}_{i=i_s}^{m})$$

where $k_f = m - i_s$. The reverse motion is then defined as:
$$\{\mathbf{R}_j^{\text{rev}}\}_{j=0}^{k_f-2} = \text{Flip}\left(\{\mathbf{R}_j^{\text{fwd}}\}_{j=0}^{k_f-2}\right)$$

Finally, we construct the symmetric update:
$$\{\mathbf{R}_i^{\text{new}}\}_{i=i_s}^{i_e-1} = \{\mathbf{R}_j^{\text{fwd}}\}_{j=0}^{k_f-1} \cup \{\mathbf{R}_j^{\text{rev}}\}_{j=0}^{k_f-2}$$

| Rotation Type | $< \delta^x_{\text{avg all}} \uparrow$ | $< \delta^x_{\text{avg if}} \uparrow$ | $< \delta^x_{\text{avg oof}} \uparrow$ | $AD^x_{\text{avg all}} \downarrow$ | $AD^x_{\text{avg if}} \downarrow$ | $AD^x_{\text{avg oof}} \downarrow$ |
|---|---|---|---|---|---|---|
| Spiral | $0.3719_{\pm 0.1359}$ | $0.5367_{\pm 0.1229}$ | $0.2733_{\pm 0.1538}$ | $3.8903_{\pm 5.0030}$ | $2.0936_{\pm 2.5373}$ | $4.9421_{\pm 6.6042}$ |
| Random | $0.1641_{\pm 0.0769}$ | $0.3300_{\pm 0.1228}$ | $0.0671_{\pm 0.0585}$ | $10.9786_{\pm 8.6354}$ | $4.3892_{\pm 5.5653}$ | $14.3985_{\pm 10.7731}$ |
| Simulated Human | $0.2623_{\pm 0.0839}$ | $0.4441_{\pm 0.1248}$ | $0.0849_{\pm 0.0746}$ | $8.2641_{\pm 6.2842}$ | $3.9029_{\pm 4.3724}$ | $12.1978_{\pm 8.6577}$ |
| btf (back-to-front) | $0.2269_{\pm 0.0822}$ | $0.4128_{\pm 0.1285}$ | $0.0568_{\pm 0.0523}$ | $10.0827_{\pm 7.2369}$ | $4.0882_{\pm 4.0901}$ | $15.1320_{\pm 10.5075}$ |
| Spin X | $0.2691_{\pm 0.0729}$ | $0.4537_{\pm 0.0879}$ | $0.1255_{\pm 0.0744}$ | $6.1869_{\pm 4.7072}$ | $3.9228_{\pm 2.5157}$ | $7.8400_{\pm 6.2102}$ |
| Spin Y | $0.1448_{\pm 0.0362}$ | $0.3411_{\pm 0.0596}$ | $0.0035_{\pm 0.0055}$ | $31.3714_{\pm 10.6858}$ | $7.3867_{\pm 4.7921}$ | $49.2281_{\pm 17.7501}$ |
| Spin Z | $0.2830_{\pm 0.0778}$ | $0.4255_{\pm 0.0913}$ | $0.1379_{\pm 0.0808}$ | $6.0208_{\pm 4.4710}$ | $4.3959_{\pm 3.3885}$ | $7.6225_{\pm 5.8348}$ |
| Static | $0.1351_{\pm 0.0529}$ | $0.3212_{\pm 0.1129}$ | $0.0207_{\pm 0.0318}$ | $16.9311_{\pm 9.3992}$ | $5.1343_{\pm 3.1230}$ | $24.0963_{\pm 12.6320}$ |

Table 4: CoTracker360 results on camera-motion-type subsets of the data.

| Category | $< \delta^x_{\text{avg all}} \uparrow$ | $< \delta^x_{\text{avg if}} \uparrow$ | $< \delta^x_{\text{avg oof}} \uparrow$ | $AD^x_{\text{avg all}} \downarrow$ | $AD^x_{\text{avg if}} \downarrow$ | $AD^x_{\text{avg oof}} \downarrow$ |
|---|---|---|---|---|---|---|
| People & Blogs | $0.2299_{\pm 0.1106}$ | $0.3970_{\pm 0.1340}$ | $0.1072_{\pm 0.1055}$ | $8.6519_{\pm 7.6875}$ | $4.1197_{\pm 4.2611}$ | $11.5161_{\pm 10.1156}$ |
| Entertainment | $0.2439_{\pm 0.1129}$ | $0.4152_{\pm 0.1309}$ | $0.1186_{\pm 0.1098}$ | $8.0247_{\pm 6.7324}$ | $3.6972_{\pm 3.3647}$ | $10.7544_{\pm 9.1642}$ |
| Gaming | $0.2187_{\pm 0.1124}$ | $0.3802_{\pm 0.1396}$ | $0.1020_{\pm 0.1002}$ | $9.9713_{\pm 9.5576}$ | $4.8707_{\pm 5.2604}$ | $13.0874_{\pm 12.5304}$ |
| Music | $0.2648_{\pm 0.1232}$ | $0.4275_{\pm 0.1403}$ | $0.1432_{\pm 0.1260}$ | $7.1626_{\pm 8.4462}$ | $3.8477_{\pm 8.2595}$ | $9.3536_{\pm 9.6667}$ |
| Autos & Vehicles | $0.2342_{\pm 0.1184}$ | $0.3840_{\pm 0.1408}$ | $0.1264_{\pm 0.1130}$ | $7.9391_{\pm 6.2584}$ | $4.1250_{\pm 3.6298}$ | $10.2975_{\pm 8.4071}$ |
| Sports | $0.2349_{\pm 0.1097}$ | $0.4011_{\pm 0.1336}$ | $0.1120_{\pm 0.1025}$ | $8.3039_{\pm 7.2686}$ | $3.9764_{\pm 3.8200}$ | $11.0010_{\pm 9.7529}$ |
| Travel & Events | $0.2339_{\pm 0.1130}$ | $0.4061_{\pm 0.1339}$ | $0.1080_{\pm 0.1084}$ | $8.7443_{\pm 7.4930}$ | $3.8804_{\pm 3.5047}$ | $11.7916_{\pm 10.1733}$ |
| Film & Animation | $0.2308_{\pm 0.1122}$ | $0.3987_{\pm 0.1345}$ | $0.1095_{\pm 0.1046}$ | $8.5133_{\pm 7.4614}$ | $3.9487_{\pm 3.5353}$ | $11.1940_{\pm 9.5001}$ |
| Science & Technology | $0.2404_{\pm 0.1168}$ | $0.4128_{\pm 0.1456}$ | $0.1180_{\pm 0.1069}$ | $8.1326_{\pm 7.0445}$ | $3.9888_{\pm 4.1551}$ | $10.7030_{\pm 8.9719}$ |
| News & Politics | $0.2492_{\pm 0.1186}$ | $0.4140_{\pm 0.1399}$ | $0.1263_{\pm 0.1144}$ | $7.7162_{\pm 8.1667}$ | $3.8633_{\pm 5.6840}$ | $10.1225_{\pm 9.9194}$ |
| Comedy | $0.2557_{\pm 0.1062}$ | $0.4149_{\pm 0.1243}$ | $0.1342_{\pm 0.1059}$ | $7.2893_{\pm 6.2056}$ | $3.7960_{\pm 3.6669}$ | $9.8020_{\pm 8.8378}$ |
| Education | $0.2522_{\pm 0.1140}$ | $0.4220_{\pm 0.1285}$ | $0.1259_{\pm 0.1133}$ | $7.6536_{\pm 6.3381}$ | $3.7259_{\pm 3.6861}$ | $10.1989_{\pm 8.4563}$ |
| Nonprofits & Activism | $0.2441_{\pm 0.1146}$ | $0.4147_{\pm 0.1359}$ | $0.1207_{\pm 0.1093}$ | $7.9101_{\pm 6.7623}$ | $3.7610_{\pm 3.3106}$ | $10.4616_{\pm 9.2909}$ |
| Howto & Style | $0.2455_{\pm 0.1063}$ | $0.4120_{\pm 0.1244}$ | $0.1249_{\pm 0.1066}$ | $6.9680_{\pm 4.8016}$ | $3.4499_{\pm 2.9274}$ | $9.3267_{\pm 6.4206}$ |
| Pets & Animals | $0.2176_{\pm 0.1107}$ | $0.3993_{\pm 0.1332}$ | $0.0962_{\pm 0.0941}$ | $9.5810_{\pm 7.1057}$ | $4.0220_{\pm 2.9383}$ | $12.5009_{\pm 8.9180}$ |

Table 5: CoTracker360 results on subsets of the dataset split by category.

**Final Sequence**   The new rotation sequence is given by:

$$\mathbf{R}_i = \begin{cases} \mathbf{R}_i^{(0)} & \text{if } i \notin [i_s, i_e - 1] \\ \mathbf{R}_i^{\text{new}} & \text{otherwise} \end{cases}$$

# D   Metrics Across Rotations and Subcategories

Table 4 presents the performance of our CoTracker360 on various camera emulation methods. The model achieves the best results with the Spiral motion, likely because objects often remain partially visible or near the frame's edge, simplifying position estimation. In contrast, the Spin Y motion is the most challenging due to its roll rotation, which requires the model to track objects as they become inverted.

Table 5 presents the performance of CoTracker360 on the video subcategories of the TAPVid360-10k dataset. The results are consistent across all categories, demonstrating that our method generalises effectively to diverse objects and environments.

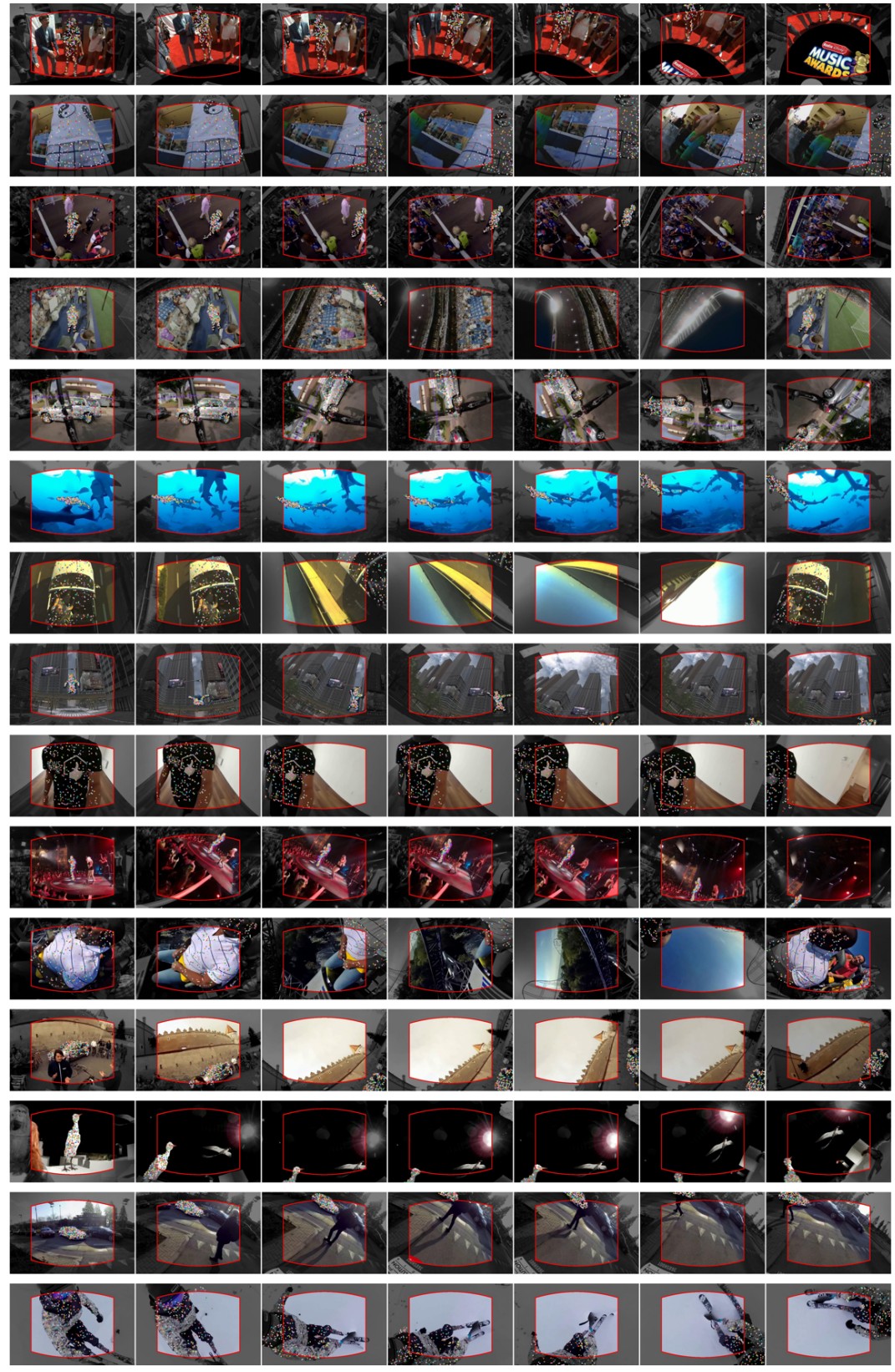

Figure 8: Further examples of the TAPVid360-10k dataset.

