# OpenReview forum: "TAPVid-360: Tracking Any Point in 360 from Narrow Field of View Video"
_NeurIPS.cc/2025/Datasets_and_Benchmarks_Track — NeurIPS 2025 Datasets and Benchmarks Track poster_

### Official Review · Reviewer_hdtJ · 2025-06-14

**Ethics Flags:** Data privacy, copyright, and consent
**Rating:** 4
**Confidence:** 5

**Summary:**

This manuscript introduces the TAPVid-360 task, the TAP360-10k dataset, and the CoTracker360 baseline for point tracking in panoramic videos, with a strong focus on out-of-frame tracking. The work addresses a critical gap in panoramic video tracking and persistent scene understanding by leveraging the unique properties of 360° videos. While the contributions are notable, issues such as ground truth noise, reliance on synthetic data, and limited discussions on real-world applications reduce the broader applicability of the work.

**Additional Feedback:**

In future work, it would be beneficial to provide the data generation pipeline scripts, as this would facilitate dataset expansion and adaptation for specific domains by other researchers.

Some beyond-the-field-of-view prediction works, such as FishDreamer, Dream360, LayerPano3D, and PanoDreamer, could be discussed in detail.

As the dataset contains various collection conditions, it would be nice to present more visualization results to analyze the performance of the proposed method and challenges in different conditions.

As a benchmark paper, it would be nice to add more comprehensive ablation and parameter studies on the benchmark, e.g., experiments for tracking using different field-of-view data.

**Dataset Code Accessibility:**

Yes

**Dataset Code Comments:**

The authors have provided the relevant test code on GitHub, as well as the corresponding dataset on Hugging Face.

**Ethical Comments:**

The dataset utilizes publicly available 360° YouTube videos (360-1M dataset), containing no personal or sensitive data. The content quality is maintained through appropriate filtering mechanisms, and the work complies with the NeurIPS Code of Ethics.

**Ethical Considerations:**

No, there are no or only very minor ethics concerns

**Final Justification:**

The reviewer would like to thank the authors for the response, which helps to solve some concerns with comparisons and benchmarks not available in the first version.

The idea of the established benchmark to track outside the field of view is interesting. Yet, the work is a preliminary effort. Many comparisons, dynamic-FoV experiments, and benchmarks of existing methods are not ready as a solid NeurIPS paper. The dataset curation is based on existing panoramic datasets, which weakens the contribution of the collected dataset. The paper would require a lot of modifications in the final version.

For these reasons, the reviewer would maintain a borderline rating.

**Limitations Weaknesses:**

1. The dataset contains annotation noise (Figure 4) that could compromise benchmark reliability. Please elaborate on the error analysis, and potential impact on results.

2. It is recommended to include comparisons with conventional 3D tracking metrics, such as 3D positional error, to better clarify the relationship and distinctions between the TAPVid-360 task and existing 3D tracking tasks.

3. The existing TAPVid-3D dataset can be made largely consistent with the task setting of this work by simply reducing its field of view. As such, the claimed relevance of the 360-degree field of view to the task appears to be not strongly substantiated.

4. The manuscript only compares the proposed method with a limited set of baselines, such as CoTracker and TAPIR. However, it lacks comparison with more recent 3D tracking approaches, such as SceneTracker.

5.  The experimental analysis would benefit from greater detail, including standard deviation of errors and tests under extreme scenarios (e.g., fast motion, wide viewpoint changes) to better evaluate robustness.

6. Typo:
- Line 125: Correct "psudeo-ground-truth" to "pseudo-ground-truth".
- The usage of "TAPVid-360" and "TapVid-360" is inconsistent.

**Strengths Contributions:**

1. Novel Task： TAPVid-360 addresses a critical gap in panoramic video tracking.
2. Dataset Contribution： The release of TAP360-10k supports further research.
3. Improved Baseline： CoTracker360 optimizes spherical geometry for 360° video tracking.
4. Panoramic Properties： Effective use of 360° video properties enhances scene understanding.

---

> ### Author Rebuttal · Authors · 2025-07-30
>
> We thank Reviewer hdtJ for their detailed and constructive feedback. The specific and actionable points raised are very valuable for improving our work, and we believe we have addressed all of their concerns below.
>
> We would like to begin by clarifying a central point about our work which, based on the reviewer's summary, we may not have communicated clearly. The TAPVid-360 task does not involve tracking points within a panoramic video at test time. Instead, the task is to track points from a conventional, narrow field-of-view (NFoV) video, with the key challenge being the prediction of temporal unit vectors, pointing towards scene points, even when they are far outside the camera's frustum. The novelty of our data generation lies in using 360° videos as a source of supervision to create NFoV videos, with out-of-frame (OOF) directional tracks. This supervision would be exceptionally difficult to acquire otherwise and is not available in existing datasets like TAPVid-3D.
>
> We believe this clarification addresses a core misunderstanding reflected in the review, particularly regarding "panoramic video tracking" (as our task is defined for narrow FoV video) and "reliance on synthetic data" (as our dataset is built exclusively from real-world 360° videos).
>
> **Evaluation of Pseudo-Ground-Truth Fidelity:**
> We thank the reviewer for this insightful comment, reviewer BmoK also highlighted this as a point of improvement, we kindly as the reviewer to see our response to them.
>
> **Comparison with 3D Positional Error Metrics:**
> The core distinction is that our TAPVid-360 task is intentionally formulated to track 3D direction (unit vectors), not absolute 3D position. This choice is central to our contribution, as it allows for a scalable data generation pipeline from ubiquitous 360° videos, circumventing the need for the complex 4D scene models required for positional tracking.
>
> Consequently, a direct comparison using 3D positional error is not feasible, as our ground truth data and task are deliberately designed to not require the depth information needed for a 3D poisitional metric. Our angular distance metrics are therefore the best way to evaluate performance on the TAPVid-360 task.
>
> We argue that this directional task is a more scalable and accessible way to learn allocentric representations. We will update the paper highlighting this distinction from traditional 3D tracking.
>
> **Relevance of 360° Data Not Substantiated:**
> TAPVid-3D is already a narrow field of view dataset. Further cropping it such that some tracked points leave the field of view would lead to unrealistically narrow field of view input videos. But more importantly, the angle of tracked points outside the field of view would be very small. By using 360 videos as our source, we can include tracked points with very large rotations outside the field of view, potentially upto 180 degrees from the original view direction. This is much more challenging for evaluation and more strongly tests the ability of models to maintain stable point tracks over large changes in view direction.
>
> **Limited Baselines e.g. SceneTracker:**
> We thank the reviewer for bringing SceneTracker and the Long-term Scene Flow Estimation (LSFE) task to our attention. We will include SceneTracker in our related work as an additional formulation of the tracking problem. However, a direct evaluation of SceneTracker is not possible. The LSFE task, as defined by SceneTracker, assumes RGB-D video as input, requiring a depth map for every frame. In contrast, our TAPVid-360 task is designed for predictions exlusivily from RGB input videos. A core component of SceneTracker is its "depth residual feature," which directly uses the input depth maps to refine its estimates. Since our dataset does not provide ground-truth depth this makes running SceneTracker not possible.
>
> We do agree however with the point raised that more comparions with TAP3D style methods is important. We have therefore added TAPIP3D [1] to the comparison methods. This was done by transforming the predicted temporal point cloud into camera space and normalising to unit vector directions. Due to the method's computational cost (~10 minutes per clip) and the limited time for this revision, our current results are based on a 1,000-clip subset. The full 10,000-clip dataset will be evaluated for the camera-ready version. As shown in Table 2, the model still struggles with out-of-frame samples, we argue this is both an architectual and lack of data issue, something our method could precisily help with, as discussed in our response to reviewer BmoK on Discussion of Downstream Uses, by using our data as pretraining for the temporal 3D tracking task.
>
> | Category | $\delta^{x}_{avg,all}$ ↑ | $\delta^{x}_{avg,if}$ ↑ | $\delta^{x}_{avg,oof}$ ↑ | $AD^{x}_{avg,all}$ ↓ | $AD^{x}_{avg,if}$ ↓ | $AD^{x}_{avg,oof}$ ↓ |
> |:---|:---:|:---:|:---:|:---:|:---:|:---:|
> | Spatial Tracker | 0.093 ± 0.077 | 0.129 ± 0.088 | 0.017 ± 0.042 | 23.788 ± 31.066 | 18.797 ± 20.677 | 33.996 ± 37.203 |
> | TAPIR | 0.010 ± 0.065 | 0.013 ± 0.078 | 0.004 ± 0.017 | 25.021 ± 33.090 | 21.011 ± 22.229 | 33.222 ± 32.759 |
> | BootsTAPIR | 0.083 ± 0.066 | 0.118 ± 0.079 | 0.012 ± 0.017 | 22.443 ± 33.192 | 17.776 ± 22.291 | 31.988 ± 32.800 |
> | CoTracker3 | 0.095 ± 0.098 | 0.131 ± 0.110 | 0.022 ± 0.059 | 21.606 ± 29.532 | 16.305 ± 17.834 | 32.449 ± 36.499 |
> | TAPNext | 0.011 ±0.062 | 0.016 ± 0.073 | 0.001 ± 0.020 | 42.600 ± 34.300 | 29.152 ± 24.049 | 70.105 ± 35.687 |
> | TAPIP3D (*1k subset*) | 0.005 ± 0.042 | 0.007 ± 0.050 | 0.001 ± 0.016 | 65.182 ± 37.731 | 55.019 ± 30.752 | 85.440 ± 41.954 |
> | *Ours* (CoTracker360) | 0.206 ± 0.281 | 0.292 ± 0.300 | 0.030 ± 0.101 | 14.992 ± 22.958 | 8.168 ± 15.255 | 28.950 ± 28.966 |
>
> **Table 1:** Additional comparions including TAPIP3D
>
> **Lack of Detailed Experimental Analysis:**
> We agree and have updated the results table to include stadard dedviation of errors, see Table 1. Below we also included evaluations on camera-motion-type subsets of the data.
>
> | Rotation Type | $<\delta^{x}_{avg,all}>$ ↑ | $<\delta^{x}_{avg,if}>$ ↑ | $<\delta^{x}_{avg,oof}>$ ↑ | $AD^{x}_{avg,all}$ ↓ | $AD^{x}_{avg,if}$ ↓ | $AD^{x}_{avg,oof}$ ↓ |
> |:---|:---:|:---:|:---:|:---:|:---:|:---:|
> | STATIC | 0.298 ± 0.308 | 0.351 ± 0.307 | 0.027 ± 0.099 | 10.318 ± 19.032 | 5.930 ± 12.212 | 32.833 ± 29.153 |
> | SPIRAL | 0.121 ± 0.237 | 0.266 ± 0.305 | 0.024 ± 0.088 | 24.537 ± 31.334 | 13.487 ± 24.402 | 31.926 ± 33.225 |
> | btf | 0.198 ± 0.281 | 0.288 ± 0.305 | 0.028 ± 0.094 | 13.599 ± 19.713 | 8.525 ± 14.708 | 23.094 ± 23.931 |
> | RANDOM | 0.122 ± 0.223 | 0.206 ± 0.265 | 0.026 ± 0.095 | 19.510 ± 23.755 | 10.592 ± 16.214 | 29.796 ± 26.729 |
> | SPIN\_X | 0.070 ± 0.185 | 0.188 ± 0.280 | 0.020 ± 0.083 | 34.978 ± 38.390 | 25.270 ± 33.886 | 39.124 ± 39.438 |
> | SPIN\_Z | 0.079 ± 0.191 | 0.186 ± 0.273 | 0.022 ± 0.085 | 30.115 ± 35.726 | 21.527 ± 32.421 | 34.665 ± 36.549 |
> | SPIN\_Y | 0.084 ± 0.197 | 0.214 ± 0.283 | 0.022 ± 0.086 | 31.688 ± 38.603 | 19.385 ± 35.168 | 37.510 ± 38.789 |
>
> **Table 2:** Results on camera-motion-type subsets of the data.
>
> **Additonal Points:**
> We agree and will be released the datageneration pipeline scripts including downloading and preprocessing of the 360° video data.
>
> We thank the reviewer for highlighting these additional works and agree a deeper discussion of other uses of 360 data is a valuable additon to the work and have extended section '360° Video for Scalable Supervision' of 'Related Work' to include them.
>
> We include below a table of results of our CoTracker360 model split across the different dataset subcategories.
>
> | Category | $<\delta^{x}_{avg,all}>$ ↑ | $<\delta^{x}_{avg,if}>$ ↑ | $<\delta^{x}_{avg,oof}>$ ↑ | $AD^{x}_{avg,all}$ ↓ | $AD^{x}_{avg,if}$ ↓ | $AD^{x}_{avg,oof}$ ↓ |
> |:---|:---:|:---:|:---:|:---:|:---:|:---:|
> | People & Blogs | 0.187 ± 0.272 | 0.271 ± 0.295 | 0.026 ± 0.093 | 16.929 ± 25.548 | 9.204 ± 17.019 | 31.858 ± 31.876 |
> | Music | 0.193 ± 0.271 | 0.275 ± 0.292 | 0.029 ± 0.098 | 14.518 ± 21.588 | 8.170 ± 14.052 | 27.240 ± 27.602 |
> | Entertainment | 0.204 ± 0.280 | 0.291 ± 0.300 | 0.030 ± 0.100 | 14.721 ± 22.430 | 8.380 ± 15.411 | 27.305 ± 28.142 |
> | Travel & Events | 0.223 ± 0.284 | 0.302 ± 0.298 | 0.033 ± 0.105 | 13.136 ± 21.117 | 7.478 ± 14.129 | 26.680 ± 27.847 |
> | Autos & Vehicles | 0.201 ± 0.275 | 0.288 ± 0.295 | 0.035 ± 0.108 | 13.253 ± 19.605 | 7.599 ± 13.706 | 23.990 ± 24.116 |
> | Film & Animation | 0.194 ± 0.284 | 0.287 ± 0.310 | 0.021 ± 0.086 | 19.465 ± 27.983 | 10.075 ± 18.321 | 36.864 ± 33.852 |
> | Gaming | 0.155 ± 0.255 | 0.242 ± 0.288 | 0.018 ± 0.078 | 21.592 ± 29.220 | 11.075 ± 19.129 | 38.300 ± 34.265 |
> | Sports | 0.180 ± 0.266 | 0.252 ± 0.290 | 0.026 ± 0.095 | 15.572 ± 22.588 | 8.864 ± 14.430 | 29.827 ± 29.210 |
> | Education | 0.203 ± 0.280 | 0.287 ± 0.299 | 0.030 ± 0.100 | 14.777 ± 22.588 | 8.281 ± 14.711 | 28.238 ± 29.163 |
>
> **Table 3:** Results on subsets of the dataset split by category
>
> We agree that extending our work to handle dynamic Fields of View (FoV) and varying camera intrinsics is an imporant next step. We acknowledged this in our limitations section (lines 297-304). Our primary goal for this initial work was to first establish the viability of the TAPVid-360 task itself. The fixed positional encoding is a limitation of the baseline, not the task. Future work could look to incorporate a per-patch directional encoding derived from the camera's intrinsic parameters. This would allow the model to reason about the FoV, making it robust to zoom. We believe this is a promising direction for future research.
>
> ### References:
>
> [1]	B. Zhang, L. Ke, A. W. Harley, and K. Fragkiadaki, ‘TAPIP3D: Tracking Any Point in Persistent 3D Geometry’, arXiv [cs.CV]. 2025.
>
> [2]	S. Koppula et al., ‘TAPVid-3D: A Benchmark for Tracking Any Point in 3D’, arXiv [cs.CV]. 2024.
>
> [3] Zholus et al., 'TAPNext: Tracking Any Point (TAP) as Next Token Prediction', arXiv [cs.CV]. 2025.

---

> > ### Comment · Reviewer_hdtJ · 2025-08-03
> > **Comment**
> >
> > The reviewer would like to thank the authors for the response, which helps to solve some concerns with comparisons and benchmarks not available in the first version.
> >
> > The idea of the established benchmark to track outside the field of view is interesting. Yet, the work is a preliminary effort. Many comparisons, dynamic-FoV experiments, and benchmarks of existing methods are not ready as a solid NeurIPS paper. The dataset curation is based on existing panoramic datasets, which limits the contribution of the dataset collection.

---

> > > ### Author Response · Authors · 2025-08-04
> > >
> > > We sincerely thank reviewer hdtJ for the continued engagement and for acknowledging the improvements from our initial rebuttal. We appreciate the opportunity to clarify the scope and significance of our contribution, particularly concerning the dataset.
> > >
> > > We believe there may be a misunderstanding of our data creation process, which we described as "curation" but is more accurately a novel data generation pipeline. The source 360-1M dataset is merely a collection of YouTube links. Our contribution is the complex, multi-stage process that transforms this raw, unstructured video into a highly structured benchmark (TAP360-10k) for the entirely new task of out-of-frame directional tracking.
> > >
> > > This involves:
> > >
> > > - Programmatically identifying and discarding unsuitable videos.
> > > - A novel pipeline, Figure. 3, that uses Lang-SAM, SAM2 and CoTracker3 to create high-fidelity ground-truth 3D directional vectors.
> > > - The generation of entirely new, narrow FoV perspective videos with controlled, dynamic camera trajectories, pairing them with the corresponding ground-truth directional vectors.
> > >
> > > The resulting NFoV, Directional Vector pairs are a completely new type of data. This asset did not exist before our work. This process allows the community to train and evaluate models on a critical aspect of allocentric scene understanding, object permanence outside the FoV, using real-world video, a task for which data was previously intractable to collect at scale.
> > >
> > > As reviewer BmoK states it “opens up a tracking paradigm with clear links to cognitive mapping and potential impact on robotics and AR” and reviewer 32HZ also sees this as “a significant contribution in itself, providing a scalable way to generate data for a task that would otherwise require difficult-to-obtain 3D annotations.”
> > >
> > > Regarding the scope, we respectfully assert that our work is on par with, and in some aspects exceeds, highly-regarded benchmarks previously accepted at NeurIPS. In terms of baselines and comparisons we have now compared against 7 other methods, including 2D and 3D methods, ensuring the latest works are included. In comparison, TAPVid-3D compared against 5 other methods (with 2 different 2D to 3D projection approaches) and TAP-Vid compared against 4 other methods. In terms of dataset sizes. Our dataset contains more clips (5,728 vs. 4,569) and more than double the number of point trajectories (5.12M vs. 2.1M) compared to TAPVid-3D (NeurIPS 2024). Furthermore our dataset is much larger than other commonly used 2D real-world TAP video datsets e.g. TAPVid-Kinetics with 1189 videos and TAPVid-DAVIS with 30 videos.
> > >
> > > Finally, while we agree that dynamic FoV is an exciting future direction, our results show that even the fixed-FoV setting poses a significant challenge for current models. We believe this benchmark strikes the right balance between feasibility and challenge, providing a clear and foundational signal for progress. The core difficulty of this task lies in reasoning about out-of-frame points, a challenge largely independent of the specific FoV.
> > >
> > > We hope this clarifies that our contribution is not the source videos, but the pipeline that creates a novel and challenging benchmark from them. We are confident that this foundational work meets the high standards of NeurIPS and kindly ask the reviewer to reconsider their assessment of its contribution.

---

> > > > ### Comment · Reviewer_hdtJ · 2025-08-04
> > > > **Comment**
> > > >
> > > > The reviewer would like to thank the authors for the clarification and update the rating to borderline acceptance.

---

### Official Review · Reviewer_32HZ · 2025-07-01

**Rating:** 4
**Confidence:** 3

**Summary:**

This paper introduces a new task, TAPVid-360, for tracking the 3D direction of any point in a video, even when the point moves far outside the camera's narrow field of view. The authors propose a scalable method to generate a large-scale dataset, TAP360-10k, by leveraging 360° videos, thus avoiding the need for complex 4D ground truth. They also present a baseline model, CoTracker360, adapted from CoTracker3, which outperforms existing methods on this new challenging task.

**Dataset Code Accessibility:**

Yes

**Dataset Code Comments:**

Authors provide enough codes and documents to use their dataset.

**Ethical Considerations:**

No, there are no or only very minor ethics concerns

**Final Justification:**

I thank the authors for their response. After careful consideration, I will be maintaining my current score.

**Limitations Weaknesses:**

1. The performance of the proposed CoTracker360 model is significantly weaker for out-of-frame points compared to in-frame points, as shown by the large gap in the AD metrics in Table 1.  While the paper acknowledges this, it lacks a deeper analysis of the potential causes. Is this performance drop a fundamental limitation of the network architecture, or is it due to an insufficient amount of training data for the model to learn the complex dynamics of out-of-view objects? Further ablation studies or analysis would strengthen the paper.


2. The qualitative results presented in Figure 4 and Figure 5 are difficult to assess due to their low resolution.  Given the dynamic nature of the task, static images are insufficient to fully appreciate the method's tracking capabilities and failure modes over time. I strongly recommend that the authors include video results in the supplementary materials to better showcase their method's performance.

**Strengths Contributions:**

1. The task formulation is novel, challenging, and significant. By shifting from 2D/2.5D tracking within the view frustum to predicting 3D directions in a panoramic context, this work addresses a key limitation of current tracking systems. This allocentric approach is a valuable step forward and is likely to attract more researchers to explore this important problem.


2. The authors have designed a sophisticated and impressive data generation pipeline to create pseudo-ground-truth labels from 360° videos. The pipeline, which intelligently combines object segmentation (Lang-SAM, SAM2), 2D tracking (CoTracker3), and novel view synthesis, is a significant contribution in itself, providing a scalable way to generate data for a task that would otherwise require difficult-to-obtain 3D annotations.

---

> ### Author Rebuttal · Authors · 2025-07-30
>
> We thank Reviewer 32HZ for their positive feedback on the novelty of the task and the impressive data generation pipeline. We are paticulaly please they also see this work as a "valuable step forward that is likely to attract more researchers to explore this important problem."
>
> **Weak Out-of-Frame Performance & Lack of Analysis:**
> We thank the reviewer for this insightful comment and agree that a deeper investigation into the cause of this gap is essential for guiding future research. To address this, we have conducted a new ablation study and provide our thoughts on the main causes.
>
> Our baseline, CoTracker360, inherits its architecture from CoTracker3, which is highly optimised for visual correspondence matching. When a point moves from in-frame (IF) to out-of-frame (OOF), the task fundamentally changes. It shifts from rich correspondence matching, which leverages strong visual signals from the CNN backbone, to a far more challenging inference problem. For OOF points, the model must infer the camera's egomotion from the visible parts of the scene and extrapolate the tracked object's own dynamics based on its history, with this complex information being maintained in the latent CNN features. We believe important architecural changes are likely required to see significant perforamance improvements. Namely, reducing inductive biases by swapping to transformer based architectures similar to TAPNext [3], and allowing a model to learn on much larger datasets now available via our data generation pipeline. And predicting probability ditributions over directions to model the inherent uncertainty once a point leaves the frame.
>
> To investigate if this problem showed signs it could be overcome simply by scaling the training data. We performed an ablation study by training CoTracker360 on datasets of varying sizes. As shown in Table 1, data is a important factor. Increasing the training data from 1k to 3k samples yields a substantial improvement in OOF performance, with the average angular distance $AD^{x}_{avg,oof}$ dropping by over 2.6°. This shows that the model can learn to better infer the ego and object motion with more examples. However, further increasing the data causes model saturation and a slight decrease in metrics performance strongly suggesting the architectural changes suggested above are required for improved performance.
>
> | Category | $<\delta^{x}_{avg,all}>$ ↑ | $<\delta^{x}_{avg,if}>$ ↑ | $<\delta^{x}_{avg,oof}>$ ↑ | $AD^{x}_{avg,all}$ ↓ | $AD^{x}_{avg,if}$ ↓ | $AD^{x}_{avg,oof}$ ↓ |
> |:---|:---:|:---:|:---:|:---:|:---:|:---:|
> | 1000 | 0.198 ± 0.280 | 0.282 ± 0.301 | 0.026 ± 0.094 | 16.703 ± 24.626 | 9.440 ± 16.854 | 31.559 ± 30.614 |
> | 3000 | 0.206 ± 0.281 | 0.292 ± 0.300 | 0.030 ± 0.101 | 14.992 ± 22.958 | 8.168 ± 15.255 | 28.950 ± 28.966 |
> | 5000 | 0.200 ± 0.277 | 0.283 ± 0.297 | 0.029 ± 0.099 | 15.136 ± 23.093 | 8.432 ± 15.382 | 28.847 ± 29.329 |
>
> **Table 1:** Ablation study on the number of training samples. We report the mean and standard deviation for each metric on the TAP360-10k test set. Increasing data from 1k to 5k samples.
>
> **Additonal Video Qualitative Results:**
> We agree this is important to both help explain the task and demonstrate failure modes of current methods. We will include video results in both the supplementary and on the dataset's project page.
>
> ### References:
>
> [1]	B. Zhang, L. Ke, A. W. Harley, and K. Fragkiadaki, ‘TAPIP3D: Tracking Any Point in Persistent 3D Geometry’, arXiv [cs.CV]. 2025.
>
> [2]	S. Koppula et al., ‘TAPVid-3D: A Benchmark for Tracking Any Point in 3D’, arXiv [cs.CV]. 2024.
>
> [3] Zholus et al., 'TAPNext: Tracking Any Point (TAP) as Next Token Prediction', arXiv [cs.CV]. 2025.

---

> > ### Comment · Reviewer_32HZ · 2025-08-04
> >
> > I thank the authors for their response. After careful consideration, I will be maintaining my current score.
> >
> > This decision is based on the following reasoning: While the problem the authors aim to address is indeed interesting, the performance of the baselines they implemented suggests that the problem is very challenging. In fact, the difficulty appears so challenging that it raises questions about whether the problem, as formulated, is fundamentally tractable.

---

> > > ### Author Response · Authors · 2025-08-04
> > >
> > > We thank the reviewer for their thoughtful consideration and agree that the task is very challenging. In fact, its difficulty is the primary motivation for our work. Many important benchmarks, such as ImageNet, initially saw very low performance, but their value was in defining an unsolved problem and providing a measure of progress. This is exactly what TAPVid-360 provides. Regarding tractability, our results already prove a strong signal exists. With only minor changes to the Co-Tracker3 architecture and a small training dataset, our baseline places 20% of points within a very small threshold window. Furthermore, as shown in our ablation study, the significant performance gain when scaling training data from 1k to 3k samples confirms the problem is learnable and the performance saturation at 5k samples shows the need for future innovation over our minor changes to Co-Tracker3.

---

### Official Review · Reviewer_BmoK · 2025-07-04

**Rating:** 5
**Confidence:** 4

**Summary:**

1. This work introduces a fresh tracking challenge—predicting the 3D bearing of any queried point relative to the camera.
2. Unlike conventional trackers that tie points to pixel coordinates within the frame, the authors represent correspondence as unit direction vectors in camera space, echoing the allocentric mapping humans perform when mentally updating object locations outside our sight.
3. They leverage 360° footage to render virtual narrow‐FOV clips, apply CoTracker3 to generate pseudo ground‐truth trajectories, and enforce quality via SAM2‐based mask consistency checks.
4.  The paper also adapts leading tracking models to this new task: after fine‐tuning and evaluating them against the proposed benchmark, the authors uncover that current state‐of‐the‐art systems still struggle with reliably estimating directions toward out‐of‐view targets.

**Dataset Code Accessibility:**

Yes

**Dataset Code Comments:**

I think that their huggingface page is clear.

**Ethical Considerations:**

No, there are no or only very minor ethics concerns

**Final Justification:**

The authors addressed my concerns during the rebuttal. I am raising my rate to 5.

**Limitations Weaknesses:**

1. The manuscript would benefit from a more explicit evaluation of pseudo‐GT fidelity perhaps via spot checks against manually annotated samples or cross‐validation with an independent 3D reconstruction.
2. While the experiments highlight that all tested methods falter on out‐of‐view points, deeper analysis identifying which architectural components or losses contribute most to the breakdown could guide future improvements.
3. A richer discussion of downstream uses.

**Strengths Contributions:**

1. Novel Task Framing. By shifting the focus from image‐based correspondence to persistent 3D direction prediction, this paper opens up a tracking paradigm with clear links to cognitive mapping and potential impact on robotics and AR.
2. Technical-sound Dataset Pipeline. They leverage 360° footage to render virtual narrow‐FOV clips, apply CoTracker3 to generate pseudo ground‐truth trajectories, and enforce quality via SAM2‐based mask consistency checks.

---

> ### Author Rebuttal · Authors · 2025-07-30
>
> We thank Reviewer BmoK for recognising our novel task framing and technically sound dataset pipeline. We are pleased they see our work "opening up a tracking paradigm with clear links to cognitive mapping and potential impact on robotics and AR". We address their comments and suggestions for improvments below.
>
> **Evaluation of Pseudo-Ground-Truth Fidelity:**
> We thank the reviewer for this insightful comment, reviewer hdtJ also highlighted this as a point of improvement and we agree that a measure of annotation noise would make the dataset and benchmark more valuable. The accuracy of using a teacher network to produce psuedo-ground-truth is an inherent challenge in creating large-scale tracking benchmarks for real-world video, where perfect ground truth is intractable. The TAPVid-3D [2] benchmark, also acknowledge this limitation stating they "inherit some limitations from the TAP and monocular depth estimation methods" used in their pipeline.
>
> While we cannot complete extensive manual verification before the rebuttal deadline, we have designed our pipeline to minimise annotation noise and below we provide a quantitative estimate of its potential impact. For the camera-ready version of our paper, we will include a dedicated section on pseudo-GT fidelity. This will include:
>
> - Manual Tracking: An analysis of fidelity based on manual annotation of a randomised subset of tracks.
>
> - Error Analysis: An analysis of common failure modes observed during these checks.
>
> Our data generation pipeline, Figure 3, is designed to use CoTracker3 in its optimal regime. For each object, we generate a perspective video where the object is kept centered and fully in-frame. This significantly reduces tracking errors that typically occur near image boundaries or during re-identification.
>
> To offer a quantitative estimate for the annotation fidelity, we can refer to the performance of CoTracker3 on existing benchmarks. On the TAP-DAVIS Benchmark, which features complex real-world motion, CoTracker3 scores 76.9 $\delta_{vis}^{avg}$ (the fraction of visible points tracked within 1, 2, 4, 8 and 16 pixels). The fraction of points within just 4, 8 and 16 pixels would be significantly higher. This is a challenging benchmark including objects being occluded and leaving frame. The performance of CoTracker3 is particularly strong when objects remain in frame. Given that our pipeline uses CoTracker3 in the "in-frame" condition, we can confidently estimate that the fidelity of our pseudo-GT is at the higher end of CoTracker3's reported performance. The primary contribution of our work is a task that encourages models to learn an allocentric understanding of the world, including object permanence for out-of-frame items. We argue this core learning signal is robust to small inaccuracies in individual tracked points. Furthermore, the scale of our dataset allows models to learn robust representations that are not overly sensitive to a small fraction of noisy samples.
>
> Finally, our data generation pipeline is modular. As point trackers improve beyond CoTracker3, our entire dataset can be regenerated to create a higher-fidelity benchmark for the community in the future.
>
> **Deeper Analysis of Failure Modes:**
> We thank the reviewer for this insightful comment and agree that a deeper investigation into the cause of this gap is essential for guiding future research. To address this, we have conducted a new ablation study and provide our thoughts on the main causes.
>
> Our baseline, CoTracker360, inherits its architecture from CoTracker3, which is highly optimised for visual correspondence matching. When a point moves from in-frame (IF) to out-of-frame (OOF), the task fundamentally changes. It shifts from rich correspondence matching, which leverages strong visual signals from the CNN backbone, to a far more challenging inference problem. For OOF points, the model must infer the camera's egomotion from the visible parts of the scene and extrapolate the tracked object's own dynamics based on its history, with this complex information being maintained in the latent CNN features. We believe important architecural changes are likely required to see significant perforamance improvements. Namely, reducing inductive biases by swapping to transformer based architectures similar to TAPNext [3], and allowing a model to learn on much larger datasets now available via our data generation pipeline. And predicting probability ditributions over directions to model the inherent uncertainty once a point leaves the frame.
>
> To investigate if this problem showed signs it could be overcome simply by scaling the training data. We performed an ablation study by training CoTracker360 on datasets of varying sizes. As shown in Table 1, data is a important factor. Increasing the training data from 1k to 3k samples yields a substantial improvement in OOF performance, with the average angular distance $AD^{x}_{avg,oof}$ dropping by over 2.6°. This shows that the model can learn to better infer the ego and object motion with more examples. However, further increasing the data causes model saturation and a slight decrease in metrics performance strongly suggesting the architectural changes suggested above are required for improved performance.
>
> | Category | $<\delta^{x}_{avg,all}>$ ↑ | $<\delta^{x}_{avg,if}>$ ↑ | $<\delta^{x}_{avg,oof}>$ ↑ | $AD^{x}_{avg,all}$ ↓ | $AD^{x}_{avg,if}$ ↓ | $AD^{x}_{avg,oof}$ ↓ |
> |:---|:---:|:---:|:---:|:---:|:---:|:---:|
> | 1000 | 0.198 ± 0.280 | 0.282 ± 0.301 | 0.026 ± 0.094 | 16.703 ± 24.626 | 9.440 ± 16.854 | 31.559 ± 30.614 |
> | 3000 | 0.206 ± 0.281 | 0.292 ± 0.300 | 0.030 ± 0.101 | 14.992 ± 22.958 | 8.168 ± 15.255 | 28.950 ± 28.966 |
> | 5000 | 0.200 ± 0.277 | 0.283 ± 0.297 | 0.029 ± 0.099 | 15.136 ± 23.093 | 8.432 ± 15.382 | 28.847 ± 29.329 |
>
> **Table 1:** Ablation study on the number of training samples. We report the mean and standard deviation for each metric on the TAP360-10k test set. Increasing data from 1k to 5k samples.
>
> **Richer Discussion of Downstream Uses:**
> We agree that this would be a valuable addition to the paper and provide the updated discussion on downstream applications below.
>
> Solving the TAPVid-360 task enables several significant downstream applications.
>
> In robotics, maintaining a persistent, panoramic understanding of its surroundings would enable more robust active vision, allowing a robot to accurately reacquire objects that leave its field-of-view (FoV) by pointing its camera in the predicted direction. The predicted directional tracks can also serve as powerful priors for re-identification (re-ID) systems. When an object reappears, candidates that appear in locations requiring implausible motion can be rejected, improving tracking robustness when objects return to view.
>
> The TAPVid-360 task requires that a model learns key cognitive abilities like object permanence and reasoning about unseen object dynamics. Given the difficulty of acquiring ground-truth temporal 3D tracks for dynamic scenes, our scalable data generation pipeline could be used to pretrain a model before being fine-tuned on smaller 3D datasets.
>
> Finally, the directional tracks could be used as a conditioning signal for video generation models. This would help enforce temporal consistency and plausible object motion, mitigating common failure modes where objects are forgotten or hallucinated incorrectly after leaving and re-entering the camera's view.
>
> ### References:
>
> [1]	B. Zhang, L. Ke, A. W. Harley, and K. Fragkiadaki, ‘TAPIP3D: Tracking Any Point in Persistent 3D Geometry’, arXiv [cs.CV]. 2025.
>
> [2]	S. Koppula et al., ‘TAPVid-3D: A Benchmark for Tracking Any Point in 3D’, arXiv [cs.CV]. 2024.
>
> [3] Zholus et al., 'TAPNext: Tracking Any Point (TAP) as Next Token Prediction', arXiv [cs.CV]. 2025.

---

> > ### Comment · Reviewer_BmoK · 2025-08-05
> >
> > Thanks to the authors for their hard work. I am willing to raise my rating to 5.

---

### Official Review · Reviewer_MMiz · 2025-07-04

**Rating:** 5
**Confidence:** 3

**Summary:**

This paper introduces TAPVid-360, a novel task for tracking any point in 360° from narrow field-of-view video. The key innovation is reformulating point tracking from 2D pixel coordinates to 3D directional vectors (unit vectors pointing from camera to scene points), enabling persistent tracking even when points move far outside the visible frame. The authors claim that their work enables learning allocentric (world-centric) scene representations that persist beyond momentary views, addressing a critical gap in current vision systems that operate primarily in an egocentric manner.

**Dataset Code Accessibility:**

Yes

**Ethical Considerations:**

No, there are no or only very minor ethics concerns

**Limitations Weaknesses:**

### 1. **Fixed Field of View Constraint**
The authors acknowledge (lines 298-302) that "The dataset is created using a fixed field of view. This allows us to test whether the TAPVid-360 task is solvable in this restricted case but does not allow us to test sensitivity to field of view or dynamically changing field of view (i.e. zoom)." This is a significant limitation because:
- Real-world applications often involve varying FOVs (zoom in/out)
- The baseline method relies on "fixed positional encoding per patch"
- Generalization to different camera intrinsics is untested

### 2. **Lack of Uncertainty Quantification**
The paper briefly mentions that "The model would also benefit from representing uncertainty - i.e. a directional distribution." This is a critical weakness because:
- Points that have been out of view for extended periods should have higher uncertainty
- The L1 loss used for supervision doesn't capture prediction confidence
- No visibility/confidence predictions are supervised during training

### 3. **Limited Diversity in Camera Motion Types**
Section C describes only 5 camera motion types (no-motion, spin, spiral, random, btf). This limited set may not represent the full complexity of real-world camera movements:
- No translation motions are included (only rotations)
- No smooth, human-like camera trajectories
- The "random" motion uses uniform distributions, which may not be realistic

**Strengths Contributions:**

### 1. **Novel Task Formulation with Clear Motivation**
The paper introduces a genuinely novel reformulation of the tracking problem, shifting from 2D pixel coordinates to 3D directional vectors. This is well-motivated by the human cognitive ability to maintain "panoramic mental models" and addresses a fundamental limitation where "most methods do not meaningfully track points once they are no longer visible". The distinction from TAP-Vid 3D is clearly articulated - while TAP-Vid 3D attempts to predict full 3D positions, TAPVid-360 focuses on directions only, making it more practical without requiring complete 3D scene reconstruction.

### 2. **Clever Dataset Generation Approach**
The methodology for generating ground truth without 3D models is innovative and scalable. By leveraging the explosion of 360° video content and using 2D tracking on full panoramas to supervise directional predictions in narrow FOV videos, the authors sidestep the expensive requirement for "Complete 3D models for a possibly dynamic scene". The pipeline in Figure 3 clearly illustrates this process.

### 3. **Comprehensive Dataset Curation**
The filtering pipeline (Section 3.1) demonstrates careful consideration of data quality, including:
- Perspective/poster detection to remove incorrectly formatted videos
- Scene dynamics filtering to ensure motion content
- Seam detection to verify true 360° content
- Watermark removal

This results in 130k high-quality 10-second clips from an initial 1M dataset.

### 4. Presentation Quality

1. **Clear Visual Communication**: Figures 1 and 3 effectively illustrate the task and methodology
2. **Well-Organized Structure**: Logical flow from motivation → dataset generation → baseline → evaluation
3. **Comprehensive Evaluation**: Separate metrics for in-frame (IF) and out-of-frame (OOF) tracking provide nuanced assessment

---

> ### Author Rebuttal · Authors · 2025-07-30
>
> We thank Reviewer MMiz for their highly positive assessment and for highlighting our paper's strengths in task formulation, dataset generation, and presentation. We are particulally pleased they see our work as a "genuinely novel reformulation of the tracking problem", that is "well-motivated" and "addresses a fundamental limitation of current work". We address their noted weaknesses below.
>
> **Fixed Field of View Constraint:**
> We agree that extending our work to handle dynamic Fields of View (FoV) and varying camera intrinsics is an imporant next step. As the reviewer notes, we acknowledged this in our limitations section (lines 297-304). Our primary goal for this initial work was to first establish the viability of the TAPVid-360 task itself. The fixed positional encoding is a limitation of the baseline, not the task. Future work could look to incorporate a per-patch directional encoding derived from the camera's intrinsic parameters. This would allow the model to reason about the FoV, making it robust to zoom. We believe this is a promising direction for future research.
>
> **Lack of Uncertainty Quantification:**
> This is again a limitation only of the baseline, not the task. But we fully agree and believe this is a critical area for improvement. For this initial baseline, we chose a simple L1 loss to demonstrate that a model could regress directional tracks. We also note that while we did not supervise the confidence outputs from the original CoTracker model, these predictions are still generated and could be used.
>
> **Limited Diversity in Camera Motion Types:**
> We thank the reviewer for raising this point, as it allows us to clarify an important detail of our data generation process. The motion types described in Appendix C are used only as rotational transformations applied when rendering narrow FoV perspective crops from 360° videos. However, the underlying 360° videos sourced from YouTube contain a wide variety of complex, real-world camera motions including both rotation and translation. Therefore, the model is required to reason about egomotion from all six degrees of freedom to solve the task.

---

### Decision · Program_Chairs · 2025-09-18

**Decision:**

Accept (poster)

**Comment:**

This paper initially received mixed but overall borderline review scores: 5, 4, 4, 3. Reviewers generally recognized the merit of this work and regard the task formulation novel, the problem well-motivated, and dataset pipeline technically solid. However, they also raised several concerns, mainly about 1) fixed FoV constraint and limited cameram motion types (MMiz); 2) the pseudo-GT fidelity (MMiz); 3) deeper analysis to identify architectural component contribution and weaknesses (BmoK and 32HZ); 4) more 3D tracking metrics and comparisons with more baselines (hdtJ).

The authors have made great efforts to address these concerns. The rebuttal was persuasive. After rebuttal, the fourth reviewer hdtJ increased the scores to 4. The second reviewer BmoK, while keeping the rating unchanged, is willing to increase the score to 5. The final ratings unanimously recommend acceptance. The AC checked the paper, rebuttal, and review comments, and recommends accepting the paper.